# Identification of Nonlinear Latent Hierarchical Models

**Lingjing Kong**[1], **Biwei Huang**[2], **Feng Xie**[3], **Eric Xing**[1,4], **Yuejie Chi**[1], and **Kun Zhang**[1,4]

[1] Carnegie Mellon University
[2] University of California San Diego
[3] Beijing Technology and Business University
[4] Mohamed bin Zayed University of Artificial Intelligence

## Abstract

Identifying latent variables and causal structures from observational data is essential to many real-world applications involving biological data, medical data, and unstructured data such as images and languages. However, this task can be highly challenging, especially when observed variables are generated by causally related latent variables and the relationships are nonlinear. In this work, we investigate the identification problem for nonlinear latent hierarchical causal models in which observed variables are generated by a set of causally related latent variables, and some latent variables may not have observed children. We show that the identifiability of causal structures and latent variables (up to invertible transformations) can be achieved under mild assumptions: on causal structures, we allow for multiple paths between any pair of variables in the graph, which relaxes latent tree assumptions in prior work; on structural functions, we permit general nonlinearity and multi-dimensional continuous variables, alleviating existing work's parametric assumptions. Specifically, we first develop an identification criterion in the form of novel identifiability guarantees for an elementary latent variable model. Leveraging this criterion, we show that *both causal structures and latent variables* of the hierarchical model can be identified asymptotically by explicitly constructing an estimation procedure. To the best of our knowledge, our work is the first to establish identifiability guarantees for both causal structures and latent variables in nonlinear latent hierarchical models.

## 1 Introduction

Classical causal structure learning algorithms often assume no latent confounders. However, it is usually impossible to enumerate and measure all task-related variables in real-world scenarios. Neglecting latent confounders may lead to spurious correlations among observed variables. Hence, much effort has been devoted to handling the confounding problem. For instance, Fast Casual Inference and its variants [Spirtes et al., 2000, Pearl, 2000, Colombo et al., 2012, Akbari et al., 2021] leverage conditional independence constraints to locate possible latent confounders and estimate causal structures among observed variables, assuming no causal relationships among latent variables. This line of approaches ends up with an equivalence class, which usually consists of many directed acyclic graphs (DAGs).

Later, several approaches have been proposed to tackle direct causal relations among latent variables with observational data. These approaches are built upon principles such as rank constraints [Silva et al., 2006a, Kummerfeld and Ramsey, 2016, Huang et al., 2022], matrix decomposition [Chandrasekaran et al., 2011, 2012, Anandkumar et al., 2013], high-order moments [Shimizu et al., 2009, Cai et al., 2019, Xie et al., 2020, Adams et al., 2021, Chen et al., 2022], copula models [Cui et al., 2018], and mixture oracles [Kivva et al., 2021]. Pearl [1988], Zhang [2004], Choi et al. [2011], Drton et al. [2017], Zhou et al. [2020], Huang et al. [2020] extend such approaches to handle tree structures

37th Conference on Neural Information Processing Systems (NeurIPS 2023).

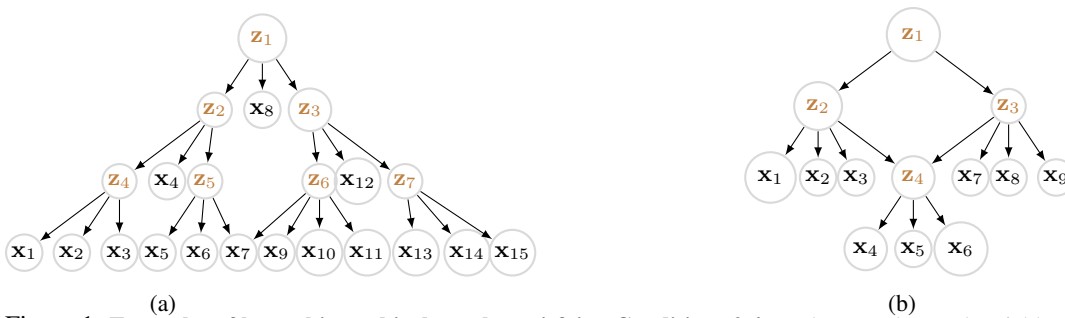

(a)              (b)

Figure 1: **Examples of latent hierarchical graphs satisfying Conditions 2.4**. $\mathbf{x}_i$ denotes observed variables and $\mathbf{z}_j$ denotes latent variables. The node size represents the dimensionality of each variable. We note that our graph condition permits multiple paths between two latent variables (e.g., $\mathbf{z}_1$ and $\mathbf{z}_4$ in Figure 1b), thus more general than tree structures.

where only one path is permitted between any pair of variables. Recently, Huang et al. [2022] propose to use rank-deficiency constraints to identify more general latent hierarchical structures. A common assumption behind these approaches is that either the causal relationships should be linear or the variables should be discrete. However, the linearity and discrete-variable assumptions are rather restrictive in real-world scenarios. Moreover, these approaches only focus on structure learning without identifying the latent variables and corresponding causal models. We defer a detailed literature review to Appendix A.

In this work, we identify the hierarchical graph structure and latent variables simultaneously for general nonlinear latent hierarchical causal models. Specifically, we first develop novel identifiability guarantees on a specific latent variable model which we refer to as the basis model (Figure 2). We then draw connections between the basis-model identifiability and the identification of nonlinear latent hierarchical models. Leveraging this connection, we develop an algorithm to localize and estimate latent variables and simultaneously learn causal structures underlying the entire system. We show the correctness of the proposed algorithm and thus obtain identifiability of both *latent hierarchical causal structures* and *latent variables* for nonlinear latent hierarchical models. In sum, our main contributions are as follows.

- Analogous to independence tests in PC algorithm [Spirtes et al., 2000] and rank-deficiency tests in Huang et al. [2022], we develop a novel identifiability theory (Theorem 3.2) as a fundamental criterion for locating and identifying latent variables in general nonlinear causal models.
- We show structure identification guarantees for latent hierarchical models admitting continuous multi-dimensional (c.f., one-dimensional and often discrete [Pearl, 1988, Choi et al., 2011, Huang et al., 2022, Xie et al., 2022]) variables, general nonlinear (c.f., linear [Pearl, 1988, Huang et al., 2022, Xie et al., 2022]) structural functions, and general graph structures (c.f., generalized trees [Pearl, 1988, Choi et al., 2011, Huang et al., 2022]).
- Under the same conditions, we provide identification guarantees for latent variables up to invertible transformations, which, to the best of our knowledge, is the first attempt in cases of nonlinear hierarchical models.
- We accompany our theory with an estimation method that can asymptotically identify the causal structure and latent variables for nonlinear latent hierarchical models and validate it on multiple synthetic and real-world datasets.

## 2 Nonlinear Latent Hierarchical Causal Models

In this section, we introduce notations and formally define the latent hierarchical causal model. We focus on causal models with directed acyclic graph (DAG) structures and denote the graph with $\mathbf{G}$, which consists of the latent variable set $\mathbf{Z} := \{\mathbf{z}_1, \ldots, \mathbf{z}_m\}$, the observed variable set [1] $\mathbf{X} := \{\mathbf{x}_1, \ldots, \mathbf{x}_n\}$, and the edge set $\mathbf{E}$. Each variable is a random vector comprising potentially multiple dimensions, i.e., $\mathbf{x}_i \in \mathbb{R}^{d_{x_i}}$ and $\mathbf{z}_j \in \mathbb{R}^{d_{z_j}}$, where $d_{x_i}$ and $d_{z_j}$ stand for the dimensionalities of $\mathbf{x}_i \in \mathbf{X}$ and $\mathbf{z}_j \in \mathbf{Z}$, respectively. Both latent variables and observed variables are generated recursively by their latent parents:

$$\mathbf{x}_i = g_{x_i}(\mathrm{Pa}(\mathbf{x}_i), \boldsymbol{\varepsilon}_{x_i}) \qquad \mathbf{z}_j = g_{z_j}(\mathrm{Pa}(\mathbf{z}_j), \boldsymbol{\varepsilon}_{z_j}), \qquad (1)$$

---

[1]We refer to leaf variables in the graph as observed variables for ease of exposition. The theory in this work is also applicable when some non-leaf variables happen to be observed, which we discuss in Append C.2

where $\text{Pa}(\mathbf{z}_j)$ represents all the parent variables of $\mathbf{z}_j$ and $\varepsilon_{z_j}$ represents the exogenous variable generating $\mathbf{z}_j$. Identical definitions apply to those notations involving $\mathbf{x}_i$. All exogenous variables $\varepsilon_{x_i}, \varepsilon_{z_j}$ are independent of each other and could also comprise multiple dimensions. We now define a general latent hierarchical causal model with a causal graph $\mathbf{G} := (\mathbf{Y}, \mathbf{E})$ in Definition 2.1.

**Definition 2.1** (Latent Hierarchical Causal Models). The variable set $\mathbf{Y}$ consists of observed variable set $\mathbf{X}$ and latent variable set $\mathbf{Z}$, and each variable is generated by Equation 1.

In light of the given definitions, we formally state the objectives of this work:

1. Structure identification: given observed variables $\mathbf{X}$, we would like to recover the edge set $\mathbf{E}$.

2. Latent-variable identification: given observed variables $\mathbf{X}$, we would like to obtain a set of variables $\hat{\mathbf{Z}} := \{\hat{\mathbf{z}}_1, \ldots, \hat{\mathbf{z}}_m\}$ where for $i \in [m]$, $\mathbf{z}_i$ and $\hat{\mathbf{z}}_i$ are identical up to an invertible mapping, i.e., $\hat{\mathbf{z}}_i = h_i(\mathbf{z}_i)$, where $h_i$ is an invertible function. [2]

Definition 2.1 gives a general class of latent hierarchical causal models. On the functional constraint, the general nonlinear function form (Equation 1) renders it highly challenging to identify either the graph structure or the latent variables. Prior work Pearl [1988], Choi et al. [2011], Xie et al. [2022], Huang et al. [2022] relies on either linear model conditions or discrete variable assumptions. In this work, we remove these constraints to address the general nonlinear case with continuous variables. On the structure constraint, identifying arbitrary causal structures is generally impossible with only observational data. For instance, tree-like structures are assumed in prior work [Pearl, 1988, Choi et al., 2011, Huang et al., 2022] for structural identifiability. In the following, we present relaxed structural conditions under which we show structural identifiability.

**Definition 2.2** (Pure Children). $\mathbf{v}_i$ is a pure child of another variable $\mathbf{v}_j$, if $\mathbf{v}_j$ is the only parent of $\mathbf{v}_i$ in the graph, i.e., $\text{Pa}(\mathbf{v}_i) = \{\mathbf{v}_j\}$.

**Definition 2.3** (Siblings). Variables $\mathbf{v}_i$ and $\mathbf{v}_j$ are siblings if $\text{Pa}(\mathbf{v}_i) \cap \text{Pa}(\mathbf{v}_j) \neq \emptyset$.

**Condition 2.4** (Structural Conditions).

   *i Each latent variable has at least 2 pure children.*
   *ii There are no directed paths between any two siblings.*

Intuitively, Condition 2.4-i allows each latent variable to leave a footprint sufficient for identification. This excludes some fundamentally unidentifiable structures. For instance, if latent variables $\mathbf{z}_1$ and $\mathbf{z}_2$ share the same children $\mathbf{X}$, i.e., $\mathbf{z}_1 \to \mathbf{X}$ and $\mathbf{z}_2 \to \mathbf{X}$, the two latent variables cannot be identified without further assumptions, while pure children would help in this case. The pure child assumption naturally holds in many applications with many directly measured variables, including psychometrics, image analysis, and natural languages. We require fewer pure children than prior work [Silva et al., 2006a, Kummerfeld and Ramsey, 2016] and place no constraints on the number of neighboring variables as in prior work [Huang et al., 2022, Xie et al., 2022].

Condition 2.4-ii avoids potential triangle structures in the latent hierarchical model. Triangles present significant obstacles for identification – in a triangle structure formed by $\mathbf{z}_1 \to \mathbf{z}_2 \to \mathbf{z}_3$ and $\mathbf{z}_1 \to \mathbf{z}_3$, it is difficult to discern how $\mathbf{z}_1$ influences $\mathbf{z}_3$ without functional constraints, as there are two directed paths between them. We defer the discussion of how to use our techniques to handle more general graphs that include triangles to Appendix C.1. Condition 2.4-ii is widely adopted in prior work, especially those on tree structures [Pearl, 1988, Choi et al., 2011, Drton et al., 2017] (which cannot handle multiple undirected paths between variables as we do) and more recent work [Huang et al., 2022]. To the best of our knowledge, only Xie et al. [2022] manage to handle triangles in the latent hierarchical structure, which, however, heavily relies on strong assumptions on both the function class (linear functions), distribution (non-Gaussian), and structures (the existence of neighboring variables).

## 3 Identifiability of Nonlinear Latent Hierarchical Causal Models

This section presents the identifiability and identification procedure of causal structures and latent variables in nonlinear latent hierarchical causal models, from only observed variables. First, in Section 3.1, we introduce a latent variable model (i.e., the *basis model* in Figure 2), whose identifiability

---

[2]We are concerned about representation vectors corresponding to each individual variable. Thus, identifiability up to invertible transforms suffices. Generally, stronger identifiability (e.g., component-wise) necessitates additional assumptions, e.g., multiple distributions [Khemakhem et al., 2020], sparsity assumptions [Zheng et al., 2022].

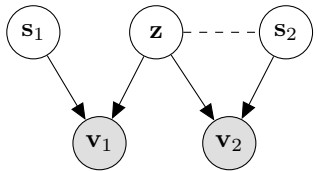

Figure 2: **The basis model**. $\mathbf{v}_1$ and $\mathbf{v}_2$ are generated from nonlinear functions $g_1$ and $g_2$ which can be distinct and non-invertible individually. We also admit dependence between $\mathbf{z}$ and $\mathbf{s}_2$, as indicated by the dashed edge.

underpins the identifiability of the hierarchical model. In Section 3.2, we establish the connection between the basis model and the hierarchical model. Leveraging this, in Section 3.3, we show by construction that both causal structures and latent variables are identifiable in the hierarchical model.

## 3.1 Basis Model Identifiability

We present the data-generating process of the basis model in Equation 2 and illustrate it in Figure 2:

$$\mathbf{v}_1 = g_1(\mathbf{z}, \mathbf{s}_1) \qquad \mathbf{v}_2 = g_2(\mathbf{z}, \mathbf{s}_2), \tag{2}$$

where $\mathbf{v}_1 \in \mathcal{V}_1 \subset \mathbb{R}^{d_{v_1}}$, $\mathbf{v}_2 \in \mathcal{V}_2 \subset \mathbb{R}^{d_{v_2}}$, $\mathbf{z} \in \mathcal{Z} \subset \mathbb{R}^{d_z}$, $\mathbf{s}_1 \in \mathcal{S}_1 \subset \mathbb{R}^{d_{s_1}}$, and $\mathbf{s}_2 \in \mathcal{S}_2 \subset \mathbb{R}^{d_{s_2}}$ are variables consisting of potentially multiple components. The data-generating process admits potential dependence between $\mathbf{z}$ and $\mathbf{s}_2$ and dependence across dimensions in each variable. We denote the entire latent variable $\mathbf{c} := [\mathbf{z}, \mathbf{s}_1, \mathbf{s}_2] \in \mathcal{C}$ and the mapping from $\mathbf{c}$ to $(\mathbf{v}_1, \mathbf{v}_2)$ as $g : \mathcal{C} \to \mathcal{V}_1 \times \mathcal{V}_2$.

Below, we introduce notations and basic definitions that we use throughout this work.
**Indexing:** For a matrix $\mathbf{M}$, we denote its $i$-th row as $\mathbf{M}_{i,:}$, its $j$-th column as $\mathbf{M}_{:,j}$, and its $(i, j)$ entry as $\mathbf{M}_{i,j}$. Similarly, for an index set $\mathcal{I} \subseteq \{1, \ldots, m\} \times \{1, \ldots, n\}$, we denote $\mathcal{I}_{i,:} := \{j : (i, j) \in \mathcal{I}\}$ and $\mathcal{I}_{:,j} := \{i : (i, j) \in \mathcal{I}\}$.
**Subspaces:** We denote a subspace of $\mathbb{R}^n$ defined by an index set $\mathcal{S}$ as $\mathbb{R}^n_{\mathcal{S}}$, where $\mathbb{R}^n_{\mathcal{S}} := \{\mathbf{z} \in \mathbb{R}^n : \forall i \notin \mathcal{S}, z_i = 0\}$.
**Support of matrix-valued functions:** We define the support of a matrix-valued function $\mathbf{M}(\mathbf{x}) : \mathcal{X} \to \mathbb{R}^{m \times n}$ as $\mathrm{Supp}(\mathbf{M}) := \{(i, j) : \exists \mathbf{x} \in \mathcal{X}, \text{s.t.}, \mathbf{M}(\mathbf{x})_{i,j} \neq 0\}$, i.e., the set of indices whose corresponding entries are nonzero for some $\mathbf{x} \in \mathcal{X}$.

In Theorem 3.2 below, we show that the latent variable $\mathbf{z}$ is identifiable up to an invertible transformation by estimating a generative model $(\hat{p}_{\mathbf{z}, \mathbf{s}_2}, \hat{p}_{\mathbf{s}_1}, \hat{g})$ according to Equation 2. We denote Jacobian matrices of $g$ and $\hat{g}$ as $\mathbf{J}_g$ and $\mathbf{J}_{\hat{g}}$, and their supports as $\mathcal{G}$ and $\hat{\mathcal{G}}$, respectively. Further, we denote as $\mathcal{T}$ a set of matrices with the same support as that of the matrix-valued function $\mathbf{J}_g^{-1}(\mathbf{c})\mathbf{J}_{\hat{g}}(\hat{\mathbf{c}})$.

**Assumption 3.1** (Basis model conditions).

   *i [Differentiability & Invertibility]: The mapping $g(\mathbf{c}) = (\mathbf{v}_1, \mathbf{v}_2)$ is a differentiable invertible function with a differentiable inverse.*

   *ii [Subspace span]: For all $i \in \{1, \ldots, d_{v_1} + d_{v_2}\}$, there exists $\{\mathbf{c}^{(\ell)}\}_{\ell=1}^{|\mathcal{G}_{i,:}|}$ and $\mathbf{T} \in \mathcal{T}$, such that $\mathrm{span}(\{\mathbf{J}_g(\mathbf{c}^{(\ell)})_{i,:}\}_{\ell=1}^{|\mathcal{G}_{i,:}|}) = \mathbb{R}^{d_c}_{\mathcal{G}_{i,:}}$ and $[\mathbf{J}_g(\mathbf{c}^{(\ell)})\mathbf{T}]_{i,:} \in \mathbb{R}^{d_c}_{\hat{\mathcal{G}}_{i,:}}$.*

   *iii [Edge Connectivity]: For all $j_z \in \{1, \ldots, d_z\}$, there exist $i_{v_1} \in \{1, \ldots, d_{v_1}\}$ and $i_{v_2} \in \{d_{v_1}, \ldots, d_{v_1} + d_{v_2}\}$, such that $(i_{v_1}, j_z) \in \mathcal{G}$ and $(i_{v_2}, j_z) \in \mathcal{G}$.*

**Theorem 3.2.** *Under Assumption 3.1, if a generative model $(\hat{p}_{\mathbf{z}, \mathbf{s}_2}, \hat{p}_{\mathbf{s}_1}, \hat{g})$ follows the data-generating process in Equation 2 and matches the true joint distribution:*

$$p_{\mathbf{v}_1, \mathbf{v}_2}(\mathbf{v}_1, \mathbf{v}_2) = \hat{p}_{\mathbf{v}_1, \mathbf{v}_2}(\mathbf{v}_1, \mathbf{v}_2), \, \forall (\mathbf{v}_1, \mathbf{v}_2) \in \mathcal{V} \times \mathcal{V}, \tag{3}$$

*then the estimated variable $\hat{\mathbf{z}}$ and the true variable $\mathbf{z}$ are equivalent up to an invertible transformation.*

The proof can be found in Appendix B.1.

**Interpretation.** Intuitively, we can infer the latent variable $\mathbf{z}$ shared by the two observed variables $\mathbf{v}_1$ and $\mathbf{v}_2$), in a sense that the estimated variable $\hat{\mathbf{z}}$ contains all the information of $\mathbf{z}$ without mixing any information from $\mathbf{s}_1$ and $\mathbf{s}_2$. The identifiability up to an invertible mapping is extensively employed in identifiability theory [Casey and Westner, 2000, Hyvärinen and Hoyer, 2000, Le et al., 2011, Theis, 2006, von Kügelgen et al., 2021, Kong et al., 2022].

**Discussion on assumptions.** Assumption 3.1-i assumes the mapping $g$ preserves latent variables' information, guaranteeing the possibility of identification. Such assumptions are universally adopted in the existing literature on nonlinear causal model identification [Hyvarinen et al., 2019, Khemakhem et al., 2020, Kong et al., 2022, Lyu et al., 2022, von Kügelgen et al., 2021].

Assumption 3.1-ii guarantees that the influence of $\mathbf{z}$ changes adequately across its domain, as discussed in prior work [Lachapelle et al., 2022, Zheng et al., 2022]. This eliminates situations where the Jacobian matrix is partially constant, causing it to insufficiently capture the connection between the observed and latent variables. This condition is specific for nonlinear functions, and a counterpart can be derived for linear cases [Zheng et al., 2022].

Assumption 3.1-iii is a formal characterization of the common cause variable z in the basis model. This assumption excludes the scenario where some dimensions of $\mathbf{z}$ only influence one of $\mathbf{v}_1$ and $\mathbf{v}_2$, in which case these dimensions of $\mathbf{z}$ are not truly the common cause of $\mathbf{v}_1$ and $\mathbf{v}_2$ and should be modeled as a separate variable rather than part of $\mathbf{z}$. This is equivalent to causal minimality, which is necessary for casual structure identification for general function classes with observational data [Peters et al., 2017].

**Comparison with prior work.** In contrast to similar data-generating processes in prior work [Lyu et al., 2022, von Kügelgen et al., 2021], Theorem 3.2 allows the most general conditions. First, Theorem 3.2 does not necessitate the invertibility for $g_1(\cdot)$ and $g_2(\cdot)$ individually as in [Lyu et al., 2022, von Kügelgen et al., 2021]. The invertibility of $g_1$ and $g_2$ amounts to asking that $\mathbf{z}$'s information is duplicate over $\mathbf{v}_1$ and $\mathbf{v}_2$. In contrast, Assumption 3.1-i only requires that $\mathbf{z}$'s information is stored in $(\mathbf{v}_1, \mathbf{v}_2)$ jointly. Moreover, generating functions $g_1(\cdot)$ and $g_2(\cdot)$ are assumed identical in [von Kügelgen et al., 2021], as opposed to being potentially distinct in Theorem 3.2. Additionally, Theorem 3.2 allows for dependence between $\mathbf{z}$ and $\mathbf{s}_2$ which is absent in [Lyu et al., 2022].

These relaxations expand the applicability of the basis model. For example, the distinction between $g_1$ and $g_2$ is indispensable in our application to the hierarchical model, as we cannot assume each latent variable generates its many children and descendants through the same function. Our proof technique is distinct from prior related work, which can be of independent interest to the community.

## 3.2 Local Identifiability of Latent Hierarchical Models

In this section, we build the connection between the basis and hierarchical models. Specifically, we show in Theorem 3.4 that a careful grouping of variables in the hierarchical model enables us to apply Theorem 3.2 to identify latent variables and their causal relations in a local scope. To this end, we modify Assumption 3.1 in basis models and obtain Assumption 3.3 for hierarchical models.

**Assumption 3.3** (Hierarchical model conditions).

   i *[Differentiability]: Structure equations in Equation 1 are differentiable.*

  ii *[Information-conservation]: Any $\mathbf{z} \in \mathbf{Z}$ and exogenous variable $\varepsilon$ can be expressed as differentiable functions of all observed variables, i.e., $\mathbf{z} = f_z(\mathbf{X})$ and $\varepsilon = f_\varepsilon(\mathbf{X})$.*

 iii *[Subspace span]: For each set $\mathbf{A}$ that d-separates all its ancestors $Anc(\mathbf{A})$ and observed variables $\mathbf{X}$, i.e., $\mathbf{X} \perp\!\!\!\perp Anc(\mathbf{A})|\mathbf{A}$, for any $\mathbf{z_A} \in Pa(\mathbf{A})$, there exists an invertible mapping from a set of ancestors of $\mathbf{A}$ containing $\mathbf{z_A}$ to the separation set $\mathbf{A}$, such that this mapping satisfies the subspace span condition (i.e., Assumption 3.1-ii).*

 iv *[Edge connectivity]: The function between each latent variable $\mathbf{z}$ and each of its children $\mathbf{z}'$ has a Jacobian $\mathbf{J}$, such that for all $j \in \{1, \ldots, d_z\}$, there exists $i \in \{1, \ldots, d_{z'}\}$, such that $(i, j)$ is in the support of $\mathbf{J}$.*

**Theorem 3.4.** *In a latent hierarchical causal model that satisfies Condition 2.4 and Assumption 3.3, we consider $\mathbf{x}_i \in \mathbf{X}$ as $\mathbf{v}_1$ and $\mathbf{X} \backslash \mathbf{v}_1$ as $\mathbf{v}_2$ in the basis model (Figure 2).* [3] *With an estimation model $(\hat{p}_{\mathbf{z}, \mathbf{s}_2}, \hat{p}_{\mathbf{s}_1}, \hat{g})$ that follows the data-generating process in Equation 2, the estimated $\hat{\mathbf{z}}$ is a one-to-one mapping of the parent(s) of $\mathbf{v}_1$, i.e., $\hat{\mathbf{z}} = h(Pa(\mathbf{v}_1))$ where $h(\cdot)$ is an invertible function.*
The proof can be found in Appendix B.2.

**Interpretation.** Theorem 3.4 shows that invoking Theorem 3.2 with the assignment $\mathbf{v}_1 = \mathbf{x}$ and $\mathbf{v}_2 = \mathbf{X} \backslash \{\mathbf{x}\}$ can identify the parents of $\mathbf{x}$ in the hierarchical model. Intuitively, we can identify latent variables one "level" above the current level $\mathbf{X}$. Moreover, as the estimated variables are equivalent to true variables up to one-to-one mappings, we take a step further in Section 3.3 to show that this procedure can be applied to the newly estimated variables iteratively to traverse the hierarchical model "level" by "level". In the degenerate case where $\mathbf{v}_1$ does not have any parents, i.e., the root of the hierarchical model, the identified variable $\mathbf{z}$ in the basis model corresponds to the $\mathbf{v}_1$ itself – we can regard $\mathbf{s}_1$ as a constant and $g_1(\cdot)$ as an identity function in Theorem 3.2.

---

[3] To avoid cluttering, we slightly abuse the bold lowercase font to represent either an individual vector or a vector set (which can be viewed as a concatenation).

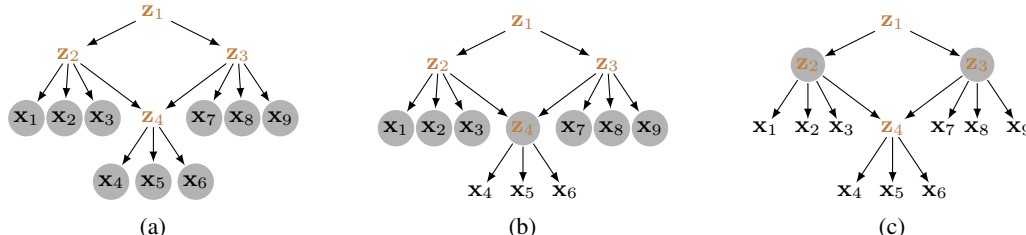

(a)  (b)  (c)

Figure 3: **Evolution of active set A in Algorithm 1.** We mark the active set $\mathbf{A}$ with shaded gray circles before each iterations of Algorithm 1, with Figure 3a, Figure 3b, and Figure 3c corresponding to iteration 1, 2, and 3. Before iteration 1, $\mathbf{A}$ is set to $\mathbf{X}$ by default. At iteration 1, $\mathbf{z}_2$, $\mathbf{z}_3$, and $\mathbf{z}_4$ can be estimated by the basis model; however, only $\mathbf{z}_4$ can be updated into $\mathbf{A}$. Otherwise, directed paths would be introduced by $\mathbf{z}_2$ and $\mathbf{z}_3$.

**Discussion on assumptions.** Assumption 3.3-ii corresponds to Assumption 3.1-i for the basis model. Intuitively, it states that the information in the system is preserved without unnecessary duplication. As far as we know, the existing literature on causal variable identification for nonlinear models [Hyvarinen et al., 2019, Khemakhem et al., 2020, Kong et al., 2022, Lyu et al., 2022, Yao et al., 2022] universally assumes the invertibility of such a mapping. Without this assumption, we cannot guarantee the identification of latent variables from observed variables in the nonlinear case.

Assumption 3.3-iii is an extension of Assumption 3.1-ii to the hierarchical case, with $\mathbf{A}$ and $\mathbf{z_A}$ playing the role of $(\mathbf{v}_1, \mathbf{v}_2)$ and $\mathbf{z}$ respectively.

Assumption 3.3-iv excludes degenerate cases where an edge connects components that do not influence each other. This is equivalent to the causal minimality and generally considered necessary for causal identification with observational data [Peters et al., 2017]. For instance, we cannot distinguish $Y := f(X) + N_Y$ and $Y := c + N_Y$ if $f$ is allowed to differ from constant $c$ only outside $X$'s support. Further, when $\mathbf{v}_1 = \mathbf{x}$ is a pure child of a latent variable $\mathbf{z}$, Condition 2.4-i ensures that $\mathbf{v}_2 := \mathbf{A_z} \setminus \{\mathbf{v}_1\}$ contains at least one pure child or descendant of $\mathbf{z}$ to fulfill Assumption 3.1-iii.

**Implications on measurement causal models.** Despite being an intermediate step towards the global structure identification, Theorem 3.4 can be applied to a class of nonlinear measurement models [4] with arbitrary latent structures and identify all the latent variables. Then, the latent structures in the measurement model can be identified by performing existing causal discovery algorithms, such as the PC algorithm [Spirtes et al., 2001], on the identified latent variables. We leave the detailed presentation of this application as future work.

### 3.3 Global Identifiability of Latent Hierarchical Causal Models

Equipped with the local identifiability (i.e., Theorem 3.4), the following theorem shows that *all* causal structures and latent variables are identifiable in the hierarchical model.

**Theorem 3.5.** *Under assumptions in Theorem 3.4, all latent variables $\mathbf{Z}$ in the hierarchical model can be identified up to one-to-one mappings, and the causal structure $\mathbf{G}$ can also be identified.*

**Comparison to prior work.** Theorem 3.5 handles general nonlinear cases, whereas prior work [Pearl, 1988, Choi et al., 2011, Huang et al., 2022, Xie et al., 2022] is limited to linear function or discrete latent variable conditions. Structure-wise, our identifiability results apply to latent structures with V-structures and certain triangle structures (see discussion in Appendix C.1) beyond generalized trees as studied in prior work [Choi et al., 2011, Pearl, 1988, Huang et al., 2022], require fewer pure children and no neighboring variables in comparison with [Xie et al., 2022, Huang et al., 2022], and can determine directions for each edge (c.f., Markov-equivalent classes in Huang et al. [2022]).

**Proof sketch with search procedures.** The proof can be found in Appendix B.3. In particular, we develop a concrete algorithm (i.e., Algorithm 1) and show that it can successfully identify the causal structure and latent variables asymptotically. We give a proof sketch of Theorem 3.4 below by navigating through Algorithm 1 and illustrate it with an example in Figure 3.

*Stage 1: identifying parents with the basis model* (line 4-line 5). As shown in Theorem 3.4, the basis model can identify the latent parents of leaf-level variables in the hierarchy. In Figure 3a, we can identify $\mathbf{z}_2$ when applying the basis model with the assignment $\mathbf{v}_1 := \{\mathbf{x}_1\}$ and $\mathbf{v}_2 := \mathbf{X} \setminus \mathbf{v}_1$. Naturally, the basic idea of Algorithm 1 is to iteratively apply the basis model to the most recently

---

[4]We refer to Silva et al. [2006b] for a general measurement model definition. Here, we consider a popular type of measurement models that has been widely used in the literature (see Xie et al. [2020]) in which observed variables do not cause other variables.

identified latent variables to further identify their parents, which is the purpose of Stage 1 (line 4-line 5). In Algorithm 1, we define as active set $\mathbf{A}$ the set of variables to which we apply the basis model. For example, $\mathbf{A}$ equals to $\mathbf{X}$ in the first round (Figure 3a) and becomes $\{\mathbf{x}_1, \mathbf{x}_2, \mathbf{x}_3, \mathbf{z}_4, \mathbf{x}_7, \mathbf{x}_8, \mathbf{x}_9\}$ in the second round (Figure 3b).

*Stage 2: merging duplicate variables* (line 6). As multiple variables in $\mathbf{A}$ can share a parent, dictionary $\mathbf{P}(\cdot)$ may contain multiple variables that are one-to-one mappings to each other, which would obscure the true causal structure and increase the algorithm's complexity. Stage 2 (Line 6) detects such duplicates and represents them with one variable. In Figure 3a, setting $\mathbf{v}_1$ to any of $\mathbf{x}_1$, $\mathbf{x}_2$, and $\mathbf{x}_3$ would yield an equivalent variable of $\mathbf{z}_2$. We merge the three equivalents by randomly selecting one.

*Stage 3: detecting and merging super-variables* (line 7 - line 9). Due to the potential existence of V-structures, variables in $\mathbf{A}$ may have multiple parents and produce super-variables in $\mathbf{P}$. For instance, at the second iteration of Algorithm 1 (i.e., Figure 3b), the basis model will be run on $\mathbf{v}_1 = \mathbf{z}_4$ and $\mathbf{v}_2 = \{\mathbf{x}_1, \mathbf{x}_2, \mathbf{x}_3, \mathbf{x}_7, \mathbf{x}_8, \mathbf{x}_9\}$ and output a variable equivalent to the concatenation $(\mathbf{z}_2, \mathbf{z}_3)$. Leaving this super-variable untouched will be problematic, as we would generate a false causal structure $\tilde{\mathbf{z}} \to \mathbf{z}_4$ where $\tilde{\mathbf{z}}$ is the estimated super-variable $(\mathbf{z}_2, \mathbf{z}_3)$, rather than recognizing $\mathbf{z}_4$ is the child of two already identified variables $\mathbf{z}_2$ and $\mathbf{z}_3$. Line 7 to line 9 detect such super-variables by testing whether each variable $\hat{\mathbf{z}}$ in $\mathbf{P}$ is equivalent to a union of other variables in $\mathbf{P}$. If such a union $\tilde{\mathbf{Z}} := \{\hat{\mathbf{z}}_1, \ldots, \hat{\mathbf{z}}_m\}$ exists, we will replace $\hat{\mathbf{z}}$ with $\tilde{\mathbf{Z}}$ in all its occurrences. In Figure 3b, we would split the variable equivalent to $[\mathbf{z}_2, \mathbf{z}_3]$ into variables of $\mathbf{z}_2$ and $\mathbf{z}_3$ individually. If $\hat{\mathbf{z}}$ is tested to be a super-variable, i.e., it can perfectly predict another variable $\hat{\mathbf{z}}'$ in $\mathbf{P}$ and $\hat{\mathbf{z}}'$ cannot predict $\hat{\mathbf{z}}$ perfectly, and the equivalent union cannot be found, we will track $\hat{\mathbf{z}}$ in line 9 to prevent it from being updated into $\mathbf{A}$ at line 16.

*Stage 4: detecting and avoiding directed paths in* $\mathbf{A}$ (line 12-line 14). Ideally, we would like to repeat line 4 to line 9 until reaching the root of the hierarchical model. Unfortunately, such an approach can be problematic, as this would cause variables in active set $\mathbf{A}$ to have directed edges among them, whereas Theorem 3.4 applies to leaf variable set $\mathbf{X}$ which contains no directed edges. In Figure 3a, $\mathbf{P}$ would contain $\{\mathbf{z}_2, \mathbf{z}_3, \mathbf{z}_4\}$, as each of these latent variables has pure observed children in $\mathbf{X}$. However, due to direct paths within $\mathbf{P}$, i.e., $\mathbf{z}_2 \to \mathbf{z}_4$ and $\mathbf{z}_3 \to \mathbf{z}_4$, we cannot directly substitute $\mathbf{X}$ with $\{\mathbf{z}_2, \mathbf{z}_3, \mathbf{z}_4\}$ in $\mathbf{A}$. To resolve this dilemma, we proactively detect directed paths emerging with newly estimated variables and defer the local update of such estimated variables to eliminate direct paths. For directed path detection, we introduce a corollary of Theorem 3.4 as Corollary 3.6.

**Corollary 3.6.** *Under assumptions in Theorem 3.4, for any $\mathbf{z} \in Pa(\mathbf{A})$ where $\mathbf{A}$ is the active set in Algorithm 1, we consider $\mathbf{v}_1 := \mathbf{z}$ and $\mathbf{v}_2 := \mathbf{A} \backslash (\tilde{C}h(\mathbf{z}) \cap \mathbf{A})$ where $\tilde{C}h(\mathbf{z})$ is a strict subset of $\mathbf{z}$'s children (i.e., when the active set $\mathbf{A}$ contains at least one child of $\mathbf{z}$'s), estimating the basis model yields $\hat{\mathbf{z}}$ equivalent to $\mathbf{z}$ up to a one-to-one mapping.*

Corollary 3.6 can be obtained by setting $\mathbf{v}_1$ as an ancestor of $\mathbf{v}_2$ and $\mathbf{s}_1$ as a degenerate variable (i.e., a deterministic quantity) in Theorem 3.4. Leveraging Corollary 3.6, we can detect whether each newly identified latent variable in $\mathbf{P}$ would introduce directed paths into $\mathbf{A}$ if they were substituted in. If directed paths exist, the variable $\hat{\mathbf{z}}$ would contain the same information as $\mathbf{v}_1 := \mathbf{z}$, which prediction tests can evaluate. In this event, we will suppress the update of the $\mathbf{z}$ at this iteration. That is, we still keep its children in $\mathbf{A}$. This directed-path detection is conducted in lines 12- 14, after properly grouping variables that share children in lines 10- 11. As shown in Figure 3b, $\mathbf{z}_2$ and $\mathbf{z}_3$ are not placed in $\mathbf{A}$, even if they are found in the first iteration. This update only happens when $\mathbf{z}_4$ has been placed in $\mathbf{A}$ at the second iteration.

Generally, a latent variable enters $\mathbf{A}$ only if all its children reside in $\mathbf{A}$. We can show that $\mathbf{A}$ contains all the information of unidentified latent variables – $\mathbf{A}$ d-separates the latent variables that have not been placed in $\mathbf{A}$ and those were in $\mathbf{A}$ once. Equipped with such a procedure, we can identify the hierarchical model iteratively until completion. We discuss Algorithm 1's complexity in Appendix D.

## 4 Experimental Results

In this section, we present experiments to corroborate our theoretical results in Section 3. We start with the problem of recovering the basis model in Section 4.2, which is the foundation of the overall identifiability. In Section 4.3, we present experiments for hierarchical models on a synthetic dataset and two real-world datasets.

**Algorithm 1** Identification of Latent Hierarchical Models. **A**: the active set, **X**: the observed variable set, **P/JointP**: dictionaries that store the relations between variables in **A** and estimated variables.

---

1: Initialize the active set: $\mathbf{A} \leftarrow \mathbf{X}$;
2: **while** $\mathbf{A} \neq \emptyset$ **do**
3:    initialize an empty set $\mathbf{R}$ and empty dictionaries **P**, **JointP**;
4:    **for** each active variable $\mathbf{a} \in \mathbf{A}$ **do**
5:       estimate the basis model with $\mathbf{v}_1 = \mathbf{a}$ and $\mathbf{v}_2 = \mathbf{A} \backslash \{\mathbf{a}\}$ to obtain $\hat{\mathbf{z}}$, and $\mathbf{P} \leftarrow \mathbf{P} \cup \{(\mathbf{a}, \hat{\mathbf{z}})\}$;
6:    merge equivalent variables in **P**;
7:    **for** each variable $\hat{\mathbf{z}}$ in **P** **do**
8:       **if** $\exists \hat{\mathbf{z}}' \in \mathbf{P}$ that $\hat{\mathbf{z}}$ can perfectly predict but $\hat{\mathbf{z}}'$ cannot predict $\hat{\mathbf{z}}$ **then**
9:          **if** $\exists \tilde{\mathbf{Z}} \subseteq \mathbf{P} \backslash \{\hat{\mathbf{z}}\}$ that perfectly predicts $\hat{\mathbf{z}}$, **then** replace $\hat{\mathbf{z}}$ with $\tilde{\mathbf{Z}}$; **else**: $\mathbf{R} \leftarrow \mathbf{R} \cup \{\hat{\mathbf{z}}\}$;
10:    cluster $\hat{\mathbf{z}}$ variables in **P** into $\{\mathbf{Z}_i\}_{i=1}^m$ such that $\mathbf{Z}_i$ is either a singleton or contains spouse variables;
11:    store each cluster $\mathbf{Z}_i$ and its children set $\mathbf{H}_i$ as a pair $(\mathbf{Z}_i, \mathbf{H}_i)$ into **JointP**;
12:    **for** each pair $(\mathbf{Z}_i, \mathbf{H}_i) \in$ **JointP** **do**
13:       estimate the basis model with $\mathbf{v}_1 = \mathbf{Z}_i$ and $\mathbf{v}_2 = \mathbf{A} \backslash \mathbf{H}_i$ to obtain a variable $\hat{\mathbf{z}}_{\text{test}}$;
14:       **if** $\exists \mathbf{z}' \in \mathbf{Z}_i$ such that $\hat{\mathbf{z}}_{\text{test}}$ can perfectly predict $\mathbf{z}'$, **then** $\mathbf{R} \leftarrow \mathbf{R} \cup \mathbf{Z}_i$
15:    **for** each pair $(\mathbf{Z}_i, \mathbf{H}_i) \in$ **JointP** **do**
16:       **if** no variables in $\mathbf{Z}_i$ has been tracked by $\mathbf{R}$, i.e., $\mathbf{Z}_i \cap \mathbf{R} = \emptyset$, **then** substitute $\mathbf{H}_i$ with $\mathbf{Z}_i$ in **A**.
17:    remove variable $\mathbf{a}$ from $\mathbf{A}$ if $\mathbf{a} \perp\!\!\!\perp \mathbf{A} \backslash \{\mathbf{a}\}$.
18: **Return**: all the past active sets **A** and parent sets **P**.

---

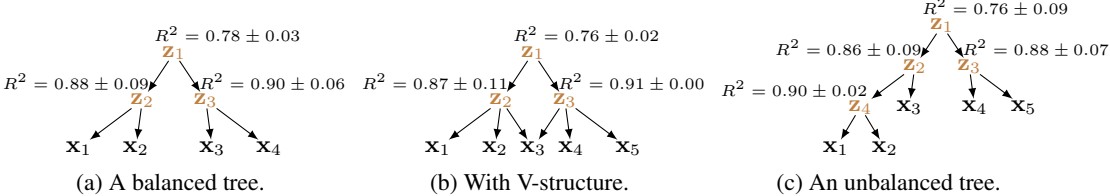

(a) A balanced tree.        (b) With V-structure.        (c) An unbalanced tree.

Figure 4: **Evaluated hierarchical models.** We denote the estimation $R^2$ scores around corresponding latent variables. We can observe that all latent variables can be identified decently, justifying our theoretic results.

### 4.1 Experimental Setup

**Synthetic data generation.** For the basis model (Figure 2), we sample $\mathbf{z} \sim \mathcal{N}(\mathbf{0}, \mathbf{I})$, $\mathbf{s}_1 \sim \mathcal{N}(\mathbf{0}, \mathbf{I})$, and $\mathbf{s}_2 \sim \mathcal{N}(\mathbf{Az} + \mathbf{b}, \mathbf{I})$, where the dependence between $\mathbf{z}$ and $\mathbf{s}_2$ is implemented by randomly constructed matrix $\mathbf{A}$ and bias $\mathbf{b}$. We choose the true mixing function $\mathbf{g}$ as a multilayer perceptron (MLP) with Leaky-ReLU activations and well-conditioned weights to facilitate invertibility. We apply element-wise $\max\{z, 0\}$ to $\mathbf{z}$ before inputting $[\mathbf{z}, \mathbf{s}_1]$ to $g_1$ and element-wise $\min\{z, 0\}$ to $\mathbf{z}$ before inputting $[\mathbf{z}, \mathbf{s}_2]$ to $g_2$, so that $\mathbf{v}_1$ is generated by the positive elements of $\mathbf{z}$ and $\mathbf{v}_2$ is generated by the negative elements of $\mathbf{z}$. This way, $g_1$ and $g_2$ are jointly invertible but not individually invertible. For latent hierarchical models (Figure 4), we sample each exogenous variable $\boldsymbol{\varepsilon}$ as $\boldsymbol{\varepsilon} \sim \mathcal{N}(\mathbf{0}, \mathbf{I})$ and each endogenous variable $\mathbf{z}$ is endowed with a distinct generating function $g_{\mathbf{z}}$ parameterized by an MLP, i.e., $\mathbf{z} = g_{\mathbf{z}}(\text{Pa}(\mathbf{z}), \boldsymbol{\varepsilon}_{\mathbf{z}})$.

**Real-world datasets.** We adopt two real-world datasets with hierarchical generating processes, namely a personality dataset and a digit dataset. The personality dataset was curated through an interactive online personality test [Project, 2019]. Participants were requested to provide a rating for each question on a five-point scale. Each question was designed to be associated with one of the five personality attributes, i.e., agreeableness, openness, conscientiousness, extraversion, and neuroticism. The corresponding answer scores are denoted as $a_i$, $o_j$, etc, as indicated in Figure 5. We use responses (around 19,500 for each question) to six questions for each of the five personality attributes. For the digit dataset, we construct a multi-view digit dataset from MNIST [Deng, 2012]. We first randomly crop each image to obtain two intermediate views and then randomly rotate each of the intermediate views independently to obtain four views. This procedure gives rise to a latent structure similar to that in Figure 4a. We feed images to a pretrained ResNet-44 [He et al., 2016] for dimensionality reduction ($28 \times 28 \rightarrow 64$) and run our algorithm on the produced features.

**Estimation models.** We implement the estimation model $(\hat{g}_1, \hat{g}_2, \hat{f})$ following Equation 2, where $\hat{f}$ can be seen as an encoder that transforms $(\mathbf{v}_1, \mathbf{v}_2)$ to the latent space and $(\hat{g}_1, \hat{g}_2)$ act as the decoders

| | $d_z = d_{s_1} = d_{s_2} = 2$ | $d_z = d_{s_1} = 2, d_{s_2} = 3$ | $d_z = d_{s_1} = d_{s_2} = 4$ | $d_z = d_{s_1} = 4, d_{s_2} = 6$ |
|---|---|---|---|---|
| Joint invertibility (Ours) | $0.93 \pm 0.09$ | $0.90 \pm 0.10$ | $0.89 \pm 0.02$ | $0.83 \pm 0.12$ |
| Individual invertibility | $0.67 \pm 0.06$ | NA | $0.67 \pm 0.09$ | NA |

Table 1: **The basis model identifiability**. We show the identifiability for $\mathbf{z}$ under various data dimensionalities $d_z$, $d_{s_1}$, and $d_{s_2}$ for $\mathbf{z}$, $\mathbf{s}_1$, $\mathbf{s}_2$. We compare our results with prior work that assumes both $g_1$ and $g_2$ are invertible individually. NA indicates that the model is not applicable when the dimensionalities of $\mathbf{s}_1$ and $\mathbf{s}_2$ differ. The results are averaged over 30 random seeds.

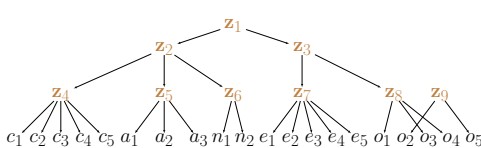

Figure 5: **The causal structure of the personality dataset learned by our method**. The Letter in each variable name indicates the personality attribute. Subscripts correspond to question indices. Some observed variables are not shown in the figure, as they are not clustered with other variables, suggesting their distant relation to the system.

generating $\mathbf{v}_1$ and $\mathbf{v}_2$ respectively. We parameterize each module with an MLP with Leaky-ReLU activation. Training configurations can be found in Appendix E.

**Evaluation metrics.** Due to the block-wise nature of our identifiability results, we adopt the coefficient of determination $R^2$ between the estimated variables $\hat{\mathbf{z}}$ and the true variables $\mathbf{z}$, where $R^2 = 1$ suggests that the estimated variable $\hat{\mathbf{z}}$ can perfectly capture the variation of the true variable $\mathbf{z}$. We apply kernel regression with Gaussian kernel to estimate the nonlinear mapping. Such a protocol is employed in related work von Kügelgen et al. [2021], Kong et al. [2022]. We repeat each experiment over at least 3 random seeds.

### 4.2 Basis Model Identification

The results for the basis model are presented in Table 1 and Figure 9. We vary the number of components of each latent partition $(d_{\mathbf{z}}, d_{\mathbf{s}_1}, d_{\mathbf{s}_2})$. We can observe that the model with individual invertibility (as assumed in prior work [von Kügelgen et al., 2021, Lyu et al., 2022]) can only capture around half of the information in $\mathbf{z}$, due to their assumption that both $g_1$ and $g_2$ are invertible, which is violated in this setup. In contrast, our estimation approach can largely recover the information of $\mathbf{z}$ across a range of latent component dimensions, verifying our Theorem 3.2. Moreover, prior work [von Kügelgen et al., 2021] assumes $g_1 = g_2$, and therefore cannot be applicable when the dimensionalities of $\mathbf{s}_1$ and $\mathbf{s}_2$ differ (e.g., $d_{s_1} = 2$, $d_{s_2} = 3$), hence the "NA" in the table. Figure 9 in Appendix E.2 shows scatter-plots of the true and the estimated components with $d_z = d_{s_1} = d_{s_2} = 2$. We can observe that components of $\mathbf{z}$ and those of $\hat{\mathbf{z}}$ are highly correlated, suggesting that the information of $\mathbf{z}$ is indeed restored. In contrast, $\hat{\mathbf{z}}$ contains very little information of $\mathbf{s}_1$, consistent with our theory that a desirable disentanglement is attainable. Additional experiments on varying sample sizes can be found in Appendix E.2.

### 4.3 Hierarchical Model Identification

**Synthetic data.** We present the evaluation of Algorithm 1 on latent hierarchical models in Figure 4, Table 2, and Table 4- 5 in Appendix E.3. In Figure 4, we can observe that all variables can be estimated decently despite a slight recovery loss from a lower to a higher level. Table 2 presents the pair-prediction scores within pairs of estimations while learning the V-structure model in Figure 4b. We can observe that scores within the sibling pairs $(\mathbf{x}_1, \mathbf{x}_2)$ and $(\mathbf{x}_4, \mathbf{x}_5)$ are much higher than non-sibling pairs. Notably, the estimate from $\mathbf{v}_1 = \mathbf{x}_3$ can perform accurate prediction over other estimates, whereas the other estimates fail to capture it faithfully. This is consistent with Theorem 3.4: the basis model with $\mathbf{v}_1 = \mathbf{x}_3$ will output a variable equivalent to the concatenation of $\mathbf{z}_2$ and $\mathbf{z}_3$, explaining the asymmetric prediction performances. These results empirically corroborate Theorem 3.5.

**Personality dataset.** From Figure 5, we can observe that our nonlinear model can correctly cluster the variables associated with the same attribute together in the first level, which is consistent with the intentions of these questions. It suggests that conscientiousness, agreeableness, and neuroticism are closely related at the intermediate level, whereas extraversion and neuroticism are remotely related. Some observed variables are not shown in the figure, as they are not clustered with other variables, indicating that they are not closely related to the system. This may provide insights into questionnaire design.

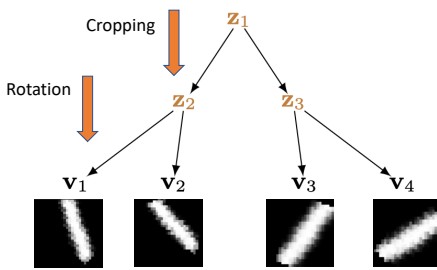

(a) **The learned causal structure.**

| | $\mathbf{v}_1$ | $\mathbf{v}_2$ | $\mathbf{v}_3$ | $\mathbf{v}_4$ |
|---|---|---|---|---|
| $\mathbf{v}_1$ | $\times$ | $\mathbf{0.69} \pm 0.001$ | $0.51 \pm 0.01$ | $0.51 \pm 0.005$ |
| $\mathbf{v}_2$ | $\mathbf{0.65} \pm 0.02$ | $\times$ | $0.48 \pm 0.01$ | $0.47 \pm 0.03$ |
| $\mathbf{v}_3$ | $0.48 \pm 0.02$ | $0.42 \pm 0.03$ | $\times$ | $\mathbf{0.78} \pm 0.05$ |
| $\mathbf{v}_4$ | $0.53 \pm 0.08$ | $0.48 \pm 0.02$ | $\mathbf{0.75} \pm 0.05$ | $\times$ |

(b) **Pairwise predictions.**

Figure 6: **Multi-view digit dataset results.** Figure 6a: $(\mathbf{v}_1, \mathbf{v}_2)$ and $(\mathbf{v}_3, \mathbf{v}_4)$ form two clusters, as they share the aspect-ratio factor within the cluster while distinct in the rotation-angle factor. Table 6b: each box $(\mathbf{a}, \mathbf{b})$ shows the $R^2$ score obtained by applying the estimated variable produced by treating one specific view $\mathbf{a}$ as $\mathbf{v}_1$ to predict the estimated variable produced by treating view $\mathbf{b}$ as $\mathbf{v}_1$.

**Digit dataset.** Figure 6a and Table 6b present the causal structure learned from the multi-view digit dataset. We can observe that we can automatically cluster the two views sharing more latent factors. This showcases that our theory and approach can handle high-dimensional variables, whereas prior causal structure learning work mostly assumes that all variables are one-dimensional.

| | $\mathbf{x}_1$ | $\mathbf{x}_2$ | $\mathbf{x}_3$ | $\mathbf{x}_4$ | $\mathbf{x}_5$ |
|---|---|---|---|---|---|
| $\mathbf{x}_1$ | $\times$ | $\mathbf{0.85} \pm 0.000$ | $0.53 \pm 0.01$ | $0.57 \pm 0.002$ | $0.55 \pm 0.003$ |
| $\mathbf{x}_2$ | $\mathbf{0.83} \pm 0.006$ | $\times$ | $0.52 \pm 0.002$ | $0.54 \pm 0.001$ | $0.53 \pm 0.000$ |
| $\mathbf{x}_3$ | $\mathbf{0.90} \pm 0.002$ | $\mathbf{0.88} \pm 0.002$ | $\times$ | $\mathbf{0.86} \pm 0.001$ | $\mathbf{0.90} \pm 0.006$ |
| $\mathbf{x}_4$ | $0.57 \pm 0.001$ | $0.54 \pm 0.002$ | $0.55 \pm 0.003$ | $\times$ | $\mathbf{0.86} \pm 0.003$ |
| $\mathbf{x}_5$ | $0.55 \pm 0.006$ | $0.56 \pm 0.001$ | $0.55 \pm 0.002$ | $\mathbf{0.83} \pm 0.003$ | $\times$ |

Table 2: **Pairwise predictions among estimated variables in Figure 4b**. Each box $(\mathbf{a}, \mathbf{b})$ shows the $R^2$ score obtained applying the estimated variable produced by treating $\mathbf{a}$ as $\mathbf{v}_1$ to predict that produced by treating $\mathbf{b}$ as $\mathbf{v}_1$. We observe that the prediction scores within sibling pairs are noticeably higher than other pairs, suggesting a decent structure estimation. In particular, the estimate from $\mathbf{v}_1 = \mathbf{x}_3$ can predict other estimates accurately, whereas not the other way round, confirming our theory that $\mathbf{v}_1 = \mathbf{x}_3$ will recover the information of both $\mathbf{z}_2$ and $\mathbf{z}_3$. The results are averaged over 30 random seeds.

## 5 Conclusion

In this work, we investigate the identifiability of causal structures and latent variables in nonlinear latent hierarchical models. We provide identifiability theory for both the causal structures and latent variables without assuming linearity/discreteness as in prior work [Pearl, 1988, Choi et al., 2011, Huang et al., 2022, Xie et al., 2022] while admitting structures more general than (generalized) latent trees [Pearl, 1988, Choi et al., 2011, Huang et al., 2022]. Together with the theory, we devise an identification algorithm and validate it across multiple synthetic and real-world datasets.

We have shown that our algorithm yields informative results across various datasets. However, it is essential to acknowledge that its primary role is as a theoretical device for our identification proof. It may not be well-suited to large-scale datasets, e.g., ImageNet, due to its nature as a discrete search algorithm. In future research, we aim to integrate our theoretical insights into scalable continuous-optimization-based algorithms [Zheng et al., 2018] and deep learning. We believe that our work facilitates the understanding of the underlying structure of highly complex and high-dimensional datasets, which is the foundation for creating more interpretable, safer, and principled machine learning systems.

**Acknowledgment.** We thank anonymous reviewers for their constructive feedback. The work of LK and YC is supported in part by NSF under the grants CCF-1901199 and DMS-2134080. This project is also partially supported by NSF Grant 2229881, the National Institutes of Health (NIH) under Contract R01HL159805, a grant from Apple Inc., a grant from KDDI Research Inc., and generous gifts from Salesforce Inc., Microsoft Research, and Amazon Research.

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

# Appendix for "Identification of Nonlinear Latent Hierarchical Models"

## A   Detailed Literature Review

Previous causal discovery approaches, which allow latent confounders and causal relationships among those latent variables, either assume linear causal relationships or assume discrete data. Representative approaches along this line include rank deficiency-based methods, matrix decomposition-based methods, generalized independent noise condition-based methods, and mixture oracles-based methods. (1) Rank deficiency-based. By testing the rank deficiency over cross-covariance matrices over observed variables, one is able to locate latent variables and identify the causal relationships among them in linear-Gaussian models. Silva et al. [2006a], Kummerfeld and Ramsey [2016] make use of the Tetrad condition, a special case of the rank-deficiency constraints, to handle the case where each observed variable is influenced by only one latent parent, and each latent variable has at least three pure measured children. The Tetrad condition has also been used to identify a latent tree structure [Pearl, 1988]. Recently, the rank-deficiency constraints have been extended to identify more general hierarchical structures [Huang et al., 2022]. (2) Matrix decomposition-based. It has been shown that, under certain conditions, the precision matrix can be decomposed into a low-rank matrix and a sparse matrix, where the low-rank matrix represents the causal structure from latent variables to observed variables and the sparse matrix gives the structural relationships over observed variables. To achieve such decomposition, certain assumptions are imposed on the structure [Chandrasekaran et al., 2011, 2012, Anandkumar et al., 2013], e.g., there should be three times more measured variables than latent variables. (3) Generalized independent noise (GIN) condition-based. The GIN condition is an extension of the independent noise condition in the existence of latent confounders, relying on higher-order statistics to identify latent structures. In particular, Xie et al. [2020] proposes a GIN-based approach that allows multiple latent parents behind every pair of observed variables and can identify causal directions among latent variables. Moreover, Adams et al. [2021] gives necessary and sufficient structural constraints in the linear, non-Gaussian or heterogeneous case, to identify the latent structures. (4) Mixture oracles-based method-based. Recently, Kivva et al. [2021] proposed a mixture oracles-based method to identify the latent variable graph that allows nonlinear causal relationships. However, it requires that the latent variables are discrete and each latent variable has measured variables as children. Thanks to the discreteness assumption, it can handle general DAGs over latent variables. On the other hand, regarding the scenario of latent hierarchical structures, most previous work along this line assumes a tree structure and requires that each variable has only one dimension and that the data is either linear-Gaussian or discrete [Pearl, 1988, Zhang, 2004, Choi et al., 2011, Drton et al., 2017, Huang et al., 2020]. In contrast, we address the general nonlinear case with continuous variables. Moreover, our conditions allow for multiple undirected paths between two variables and thus are more general than tree-based assumptions in prior work.

Another related research line is latent-variable identifiability literature. Hyvarinen et al. [2019], Khemakhem et al. [2020], Kong et al. [2022] have shown that with an additional observed variable to modulate latent independent variables, the latent independent variables are identifiable. Recently, Yao et al. [2021, 2022] allow time-delayed causal relationships among latent variables. However, for time-delayed causal relations, the causal direction is fixed and predefined, and moreover, they assume that all latent variables have measured variables as children, avoiding the hierarchical cases. Prior work[von Kügelgen et al., 2021, Lyu et al., 2022] studies latent-variable models related to our proposed basis model (which serves as a tool and is defined in Section 2) in this work, but with more restrictive functional and statistical assumptions. To the best of our knowledge, no prior work has managed to identify latent variables or causal structures in nonlinear latent hierarchical models.

We note that our work focuses on identifying latent causal models from observational data and structure conditions. Another important line of work [Ahuja et al., 2023, Squires et al., 2023, Varici et al., 2023, Jiang and Aragam, 2023, Liang et al., 2023, Buchholz et al., 2023, Zhang et al., 2023] utilizes interventional data for this purpose. Specifically, these works leverage multiple data distributions generated by one causal model under distinct interventions, which are accessible in applications like biological experiments. The accessibility of interventional data can allow for relaxed structure conditions. Hence, one should consider this tradeoff when faced with causal identification problems.

# B  Proofs

## B.1  Proof for Theorem 3.2

The original assumptions and theorem are replicated here for reference.

In Theorem 3.2, we show that the latent variable $\mathbf{z}$ shared by $\mathbf{v}_1$ and $\mathbf{v}_2$ is identifiable up to a one-to-one mapping by estimating a generative model $(\hat{p}_{\mathbf{z},\mathbf{s}_2}, \hat{p}_{\mathbf{s}_1}, \hat{g})$ according to Equation 2. We denote the support of Jacobian matrix $\mathbf{J}_g$ and $\mathbf{J}_{\hat{g}}$ as $\mathcal{G}$ and $\hat{\mathcal{G}}$ respectively and denote by $\mathbf{T}$ a matrix with the same support as $\mathbf{T}(\mathbf{c})$ in $\mathbf{J}_{\hat{g}}(\hat{\mathbf{c}}) = \mathbf{J}_g(\mathbf{c})\mathbf{T}(\mathbf{c})$.

**Assumption 3.1** (Basis model conditions)**.**

i *[Differentiability & Invertibility]: The mapping $g(\mathbf{c}) = (\mathbf{v}_1, \mathbf{v}_2)$ is a differentiable invertible function with a differentiable inverse.*

ii *[Subspace span]: For all $i \in \{1, \ldots, d_{v_1} + d_{v_2}\}$, there exists $\{\mathbf{c}^{(\ell)}\}_{\ell=1}^{|\mathcal{G}_{i,:}|}$ and $\mathbf{T} \in \mathcal{T}$, such that $span(\{\mathbf{J}_g(\mathbf{c}^{(\ell)})_{i,:}\}_{\ell=1}^{|\mathcal{G}_{i,:}|}) = \mathbb{R}^{d_c}_{\mathcal{G}_{i,:}}$ and $[\mathbf{J}_g(\mathbf{c}^{(\ell)})\mathbf{T}]_{i,:} \in \mathbb{R}^{d_c}_{\hat{\mathcal{G}}_{i,:}}$.*

iii *[Edge Connectivity]: For all $j_z \in \{1, \ldots, d_z\}$, there exist $i_{v_1} \in \{1, \ldots, d_{v_1}\}$ and $i_{v_2} \in \{d_{v_1}, \ldots, d_{v_1} + d_{v_2}\}$, such that $(i_{v_1}, j_z) \in \mathcal{G}$ and $(i_{v_2}, j_z) \in \mathcal{G}$.*

**Theorem 3.2.** *Under Assumption 3.1, if a generative model $(\hat{p}_{\mathbf{z},\mathbf{s}_2}, \hat{p}_{\mathbf{s}_1}, \hat{g})$ follows the data-generating process in Equation 2 and matches the true joint distribution:*

$$p_{\mathbf{v}_1,\mathbf{v}_2}(\mathbf{v}_1, \mathbf{v}_2) = \hat{p}_{\mathbf{v}_1,\mathbf{v}_2}(\mathbf{v}_1, \mathbf{v}_2), \ \forall (\mathbf{v}_1, \mathbf{v}_2) \in \mathcal{V} \times \mathcal{V}, \tag{3}$$

*then the estimated variable $\hat{\mathbf{z}}$ and the true variable $\mathbf{z}$ are equivalent up to an invertible transformation.*

*Proof.* We first define the indeterminacy function:

$$h := \hat{g}^{-1} \circ g,$$

which is a smooth and invertible function $h : \mathcal{C} \to \hat{\mathcal{C}}$ thanks to Assumption 3.1-i. According to the chain rule and inverse function theorem, we have the following relation among the Jacobian matrices:

$$\mathbf{J}_{\hat{g}}(\hat{\mathbf{c}}) = \mathbf{J}_g(\mathbf{c}) \cdot \mathbf{J}_h^{-1}(\mathbf{c}). \tag{4}$$

For ease of exposition, we denote $\mathbf{M}(\mathbf{c}) := \mathbf{J}_h^{-1}(\mathbf{c})$ in the following.

We define the support notations as follows:

$$\mathcal{G} := \mathrm{supp}(\mathbf{J}_g),$$
$$\hat{\mathcal{G}} := \mathrm{supp}(\mathbf{J}_{\hat{g}}),$$
$$\mathcal{T} := \mathrm{supp}(\mathbf{M}).$$

Because of Assumption 3.1-ii, for any $i \in \{1, \ldots, d_{v_1} + d_{v_2}\}$, there exists $\{\mathbf{c}^{(\ell)}\}_{\ell=1}^{|\mathcal{G}_{i,:}|}$, such that $span(\{\mathbf{J}_g(\mathbf{c}^{(\ell)})_{i,:}\}_{\ell=1}^{|\mathcal{G}_{i,:}|}) = \mathbb{R}^{d_z}_{\mathcal{G}_{i,:}}$.

Since $\{\mathbf{J}_g(\mathbf{c}^{(\ell)})_{i,:}\}_{\ell=1}^{|\mathcal{G}_{i,:}|}$ forms a basis of $\mathbb{R}^{d_c}_{\mathcal{G}_{i,:}}$, for any $j_0 \in \mathcal{G}_{i,:}$, we can express canonical basis vector $\mathbf{e}_{j_0} \in \mathbb{R}^{d_c}_{\mathcal{G}_{i,:}}$ as:

$$\mathbf{e}_{j_0} = \sum_{\ell \in \mathcal{G}_{i,:}} \alpha_\ell \cdot \mathbf{J}_g(\mathbf{c}^{(\ell)})_{i,:}, \tag{5}$$

where $\alpha_\ell \in \mathbb{R}$ is a coefficient.

Then, following Assumption 3.1-ii, there exists a deterministic matrix $\mathbf{T}$ such that

$$\mathbf{T}_{j_0,:} = \mathbf{e}_{j_0}^\top \mathbf{T} = \sum_{\ell \in \mathcal{G}_{i,:}} \alpha_\ell \cdot \mathbf{J}_g(\mathbf{c}^{(\ell)})_{i,:}\mathbf{T} \in \mathbb{R}^{d_c}_{\hat{\mathcal{G}}_{i,:}}, \tag{6}$$

where $\in$ is due to the fact that each element in the summation belongs to $\mathbb{R}^{d_c}_{\hat{\mathcal{G}}_{i,:}}$.

Therefore,

$$\forall j \in \mathcal{G}_{i,:}, \mathbf{T}_{j,:} \in \mathbb{R}^{d_c}_{\hat{\mathcal{G}}_{i,:}} .$$

Equivalently, we have:

$$\forall (i,j) \in \mathcal{G}, \quad \{i\} \times \mathcal{T}_{j,:} \subset \hat{\mathcal{G}}. \tag{7}$$

We would like to show that $\hat{\mathbf{z}}$ is not influenced by $\mathbf{s}_1$ and $\mathbf{s}_2$, which is equivalent to $\mathbf{T}_{j_z,j_{\hat{s}}} = 0$ for $j_z \in \{1, \ldots, d_z\}$ and $j_{\hat{s}} \in \{d_z + 1, \ldots, d_c\}$.

We first will prove this for $j_{\hat{s}} \in \{d_z + 1, \ldots, d_z + d_{s_1}\}$ by contradiction. Suppose that $\exists (j_z, j_{\hat{s}_1}) \in \mathcal{T}$ with $j_z \in \{1, \ldots, d_z\}$ and $j_{\hat{s}_1} \in \{d_z + 1, \ldots, d_z + d_{s_1}\}$.

Thanks to Assumption 3.1-iii, there must exist $i_{v_2} \in \{d_{v_1} + 1, \ldots, d_{v_1} + d_{v_2}\}$, such that, $(i_{v_2}, j_z) \in \mathcal{G}$.

It follows from Equation 7 that:

$$\{i_{v_2}\} \times \mathcal{T}_{j_z,:} \subset \hat{\mathcal{G}} \implies (i_{v_2}, j_{\hat{s}_1}) \in \hat{\mathcal{G}}. \tag{8}$$

However, due to the structure of $\hat{g}_2$, $[\mathbf{J}_{\hat{g}_2}]_{i_{v_2}, j_{\hat{s}_1}} = 0$, which results in a contradiction. Therefore, such $(i_{v_2}, j_{\hat{s}_1})$ does not exist. The same reasoning gives rise to that $(j_z, j_{\hat{s}_2}) \notin \mathcal{T}$ with $j_z \in \{1, \ldots, d_z\}$ and $j_{\hat{s}_2} \in \{d_z + d_{s_1} + 1, \ldots, d_c\}$. Therefore, we have shown that $\mathbf{T}_{j_z,j_{\hat{s}}} = 0$ for $j_z \in \{1, \ldots, d_z\}$ and $j_{\hat{s}} \in \{d_z + 1, \ldots, d_c\}$. As $\mathcal{T}$ is invertible, it follows from the block-matrix inverse formulae that $\mathbf{T}_{j_{\hat{z}},j_s} = 0$ for $j_{\hat{z}} \in \{1, \ldots, d_z\}$ and $j_s \in \{d_z + 1, \ldots, d_c\}$ In conclusion, $\hat{\mathbf{z}}$ is not influenced by $(\mathbf{s}_1, \mathbf{s}_2)$. Thus, we have shown that there is a one-to-one mapping between $\mathbf{z}$ and $\hat{\mathbf{z}}$. $\qquad\square$

### B.2 Proof for Theorem 3.4

The original theorem is copied below.

**Assumption 3.3** (Hierarchical model conditions)**.**

   *i [Differentiability]: Structure equations in Equation 1 are differentiable.*

   *ii [Information-conservation]: Any $\mathbf{z} \in \mathbf{Z}$ and exogenous variable $\varepsilon$ can be expressed as differentiable functions of all observed variables, i.e., $\mathbf{z} = f_z(\mathbf{X})$ and $\varepsilon = f_\varepsilon(\mathbf{X})$.*

   *iii [Subspace span]: For each set $\mathbf{A}$ that d-separates all its ancestors $Anc(\mathbf{A})$ and observed variables $\mathbf{X}$, i.e., $\mathbf{X} \perp\!\!\!\perp Anc(\mathbf{A})|\mathbf{A}$, for any $\mathbf{z}_\mathbf{A} \in Pa(\mathbf{A})$, there exists an invertible mapping from a set of ancestors of $\mathbf{A}$ containing $\mathbf{z}_\mathbf{A}$ to the separation set $\mathbf{A}$, such that this mapping satisfies the subspace span condition (i.e., Assumption 3.1-ii).*

   *iv [Edge connectivity]: The function between each latent variable $\mathbf{z}$ and each of its children $\mathbf{z}'$ has a Jacobian $\mathbf{J}$, such that for all $j \in \{1, \ldots, d_z\}$, there exists $i \in \{1, \ldots, d_{z'}\}$, such that $(i, j)$ is in the support of $\mathbf{J}$.*

**Theorem 3.4.** *In a latent hierarchical causal model that satisfies Condition 2.4 and Assumption 3.3, we consider $\mathbf{x}_i \in \mathbf{X}$ as $\mathbf{v}_1$ and $\mathbf{X}\backslash\mathbf{v}_1$ as $\mathbf{v}_2$ in the basis model (Figure 2).* [5] *With an estimation model $(\hat{p}_{\mathbf{z},\mathbf{s}_2}, \hat{p}_{\mathbf{s}_1}, \hat{g})$ that follows the data-generating process in Equation 2, the estimated $\hat{\mathbf{z}}$ is a one-to-one mapping of the parent(s) of $\mathbf{v}_1$, i.e., $\hat{\mathbf{z}} = h(Pa(\mathbf{v}_1))$ where $h(\cdot)$ is an invertible function.*

*Proof.* We show that performing estimation following the generating process Equation 2 to any $\mathbf{v}_1 = \mathbf{x}_i \in \mathbf{X}$ and $\mathbf{v}_2 = \mathbf{X} \setminus \mathbf{v}_1$ is equivalent to the estimation of the model in Figure 2 with $\mathbf{z} = Pa(\mathbf{v}_1)$. Therefore, the identifiability result ensues, thanks to Theorem 3.2.

First, we show that for each selection of $\mathbf{v}_1$, we can locate $(\mathbf{z}, \mathbf{s}_1, \mathbf{s}_2)$ such that the conditions for the basis model in Theorem 3.2 are satisfied. We choose $(\mathbf{z}, \mathbf{s}_1, \mathbf{s}_2)$ as follows.

- We choose the $\mathbf{z}$ to be all parents of $\mathbf{v}_1$: $\mathbf{z} := Pa(\mathbf{v}_1)$

- We choose $\mathbf{s}_1$ to be exogenous variables that cause $\mathbf{v}_1$ together with $\mathbf{z}$: $\mathbf{s}_1 := \varepsilon_{\mathbf{v}_1}$.

---

[5]To avoid cluttering, we slightly abuse the bold lowercase font to represent either an individual vector or a vector set (which can be viewed as a concatenation).

- To obtain $\mathbf{s}_2$, we trace back (traversing backward every edge) from the $\mathbf{v}_2$ recursively and execute the following steps. At each step of backtracking, we include any cause (exogenous variables and endogenous variables) that is not situated on any directed path from $\mathbf{z}$ to $\mathbf{v}_2$. The backtracking for each path halts when the most recent step recovers an endogenous variable out of the directed paths, or it reaches $\mathbf{z}$. We note that such a choice for $\mathbf{s}_2$ is not unique – the above procedure is only one instance.

**Invertibility.** From $(\mathbf{z}, \mathbf{s}_1, \mathbf{s}_2)$ to $(\mathbf{v}_1, \mathbf{v}_2)$, we can observe that $(\mathbf{z}, \mathbf{s}_1, \mathbf{s}_2)$ by construction include all the information to generate $(\mathbf{v}_1, \mathbf{v}_2)$, i.e., $\mathbf{X}$, as $(\mathbf{z}, \mathbf{s}_1, \mathbf{s}_2)$ d-separate their parents/ancestors from $(\mathbf{v}_1, \mathbf{v}_2)$ and contain all necessary exogenous variables. From $(\mathbf{v}_1, \mathbf{v}_2)$ to $(\mathbf{z}, \mathbf{s}_1, \mathbf{s}_2)$, we can observe that as the tuple $(\mathbf{v}_1, \mathbf{v}_2)$ comprises the entire observable set $\mathbf{X}$, which contains the information of all latent exogenous and endogenous variables according to the general invertibility (Assumption 3.3-ii) of the hierarchical model. Therefore, the mapping is invertible.

**Conditional independence: $\mathbf{v}_1 \perp\!\!\!\perp \mathbf{v}_2 | \mathbf{z}$.** As $\mathbf{z}$ is chosen to be the parents of $\mathbf{v}_1$ and there is no edge among the observables $\mathbf{X}$ (i.e., leaf variables), the local Markov property implies conditional independence.

**Subspace span.** As $(\mathbf{v}_1, \mathbf{v}_2) := \mathbf{X}$ satisfies the tree conditions in Assumption 3.3-iii w.r.t., $\mathbf{z} :=$ Pa$(\mathbf{z})$, the subspace span condition follows from Assumption 3.3-iii.

**Edge connectivity.** We assume that in the hierarchical model, the function between each latent variable $\mathbf{z}$ and each of its pure child $\mathbf{z}'$ has a non-degenerate Jacobian matrix in a sense that for all $j \in \{1, \ldots, d_z\}$, there exists $i \in \{1, \ldots, d_{z'}\}$, such that $(i, j)$ is included in the Jacobian matrix's support (i.e., Assumption 3.3-iv). Therefore, since each latent variable has at least 2 pure children (i.e., Condition 2.4-i) and $\mathbf{v}_1$ contains 1 variable, at least 1 pure child of $\mathbf{z}$ or a descendant of a pure child of $\mathbf{z}$ will appear in $\mathbf{v}_2$. Thus, for each $j \in \{1, \ldots, d_c\}$, there exist $i_{v_1}$ and $i_{v_2}$ such that both $(i_{v_1}, j)$ and $(i_{v_2}, j)$ are contained in the support of the Jacobian matrix, which fulfills the edge connectivity condition (i.e., Assumption 3.1-iii).

Thus, it follows from Theorem 3.2 that the estimated variable $\hat{\mathbf{z}}$ is a one-to-one mapping to the true variable $\mathbf{z}$.

$\square$

## B.3 Proof for Theorem 3.5

**Theorem 3.5.** *Under assumptions in Theorem 3.4, all latent variables $\mathbf{Z}$ in the hierarchical model can be identified up to one-to-one mappings, and the causal structure $\mathbf{G}$ can also be identified.*

*Proof.* We will show by induction that each iteration of Algorithm 1 fulfills the following conditions:

**Condition B.1** (Active-set conditions).

  i *Each element in the active set $\mathbf{A}$ is a one-to-one mapping of a distinct variable in $\mathbf{Y}$.*

  ii *There are no directed paths among variables in $\mathbf{A}$.*

  iii *The graph is d-separated by $\mathbf{A}$ into latent variables $\mathbf{Z_A}$ that have not been included in $\mathbf{A}$ and those $\mathbf{Z_{\bar{A}}}$ that were in $\mathbf{A}$ (but not now): $\mathbf{Z_A} \perp\!\!\!\perp \mathbf{Z_{\bar{A}}} | \mathbf{A}$.*

We will first verify the base case where active set $\mathbf{A}$ is assigned observable set $\mathbf{X}$. Condition B.1-i is automatically satisfied due to the initial assignment. As $\mathbf{X}$ are all leaf variables, there are no directed paths within $\mathbf{X}$, and therefore Condition B.1-ii is satisfied. Condition B.1-iii is also met trivially, as $\mathbf{Z_{\bar{A}}} = \emptyset$ at the initial step. So far, we have verified the base case.

Now, we make the inductive hypothesis that all conditions hold before an iteration and show that these conditions are maintained after the iteration.

We first note that Condition B.1-i, Condition B.1-ii, and Condition B.1-iii in the inductive hypothesis enable us to apply Theorem 3.4 to the modified graph consisting of variables in $\mathbf{A}$ as the bottom layer

and all latent variables that have not been placed in $\mathbf{A}$. This modified graph satisfies the structural properties required by Theorem 3.4:

- All paths end at active (observed) variables.

- There are no directed paths among the active variables.

We now analyze the updates to $\mathbf{A}$ made at Step 16 in Algorithm 1 after each iteration. For a specific variable $\mathbf{z} \in \mathbf{P}(\mathbf{A})$, there are the following cases.

**Case B.2** (One-iteration cases)**.**

  i  $\mathbf{z}$ *has only pure children and all of them are in* $\mathbf{A}$.

  ii  $\mathbf{z}$ *has only pure children and not all of them are in* $\mathbf{A}$.

  iii  $\mathbf{z}$ *has coparents and all its children and its co-parents' children are in* $\mathbf{A}$.

  iv  $\mathbf{z}$ *has coparents and not all its children and its co-parents' children are in* $\mathbf{A}$.

For Case B.2-i, when each pure child $\mathbf{a}$ of $\mathbf{z}$ is treated as $\mathbf{v}_1$, the basis model will yield $\mathbf{z}$. So, there are certainly estimates of $\mathbf{z}$ in $\mathbf{P}(\mathbf{a})$. As $\mathbf{z}$ possesses at least 2 pure children, all of which belong to $\mathbf{A}$, there will be duplicates, i.e., multiple equivalent one-to-one mappings of $\mathbf{z}$, which we merge into one at Step 6. Therefore, for each pure child $\mathbf{a}$ of $\mathbf{z}$, the only element in $\mathbf{P}(\mathbf{a})$ is a identical one-to-one mapping of $\mathbf{z}$. It follows that **JointP** contains a pair $(\mathbf{Z}, \mathbf{H})$ where $\mathbf{Z}$ consists of $\mathbf{z}$ alone and $\mathbf{H}$ contains all its pure children. This fact, together with Condition B.1-iii, implies that $\hat{\mathbf{z}}_{\text{test}}$ would be the parent of $\mathbf{z}$, which cannot perfectly predict $\mathbf{z}$ at Step 14. Therefore, Algorithm 1 will substitute all the pure children of $\mathbf{z}$ with the estimate of $\mathbf{z}$ in $\mathbf{A}$.

We now switch to Case B.2-ii. Analogous to the process in Case B.2-i, after Step 6 we will have $\mathbf{P}(\mathbf{a}) = \{\mathbf{z}\}$ for each active pure child. However, as not all pure children of $\mathbf{z}$ are active, **JointP**$(\{\mathbf{z}\})$ does not include all of the pure children of $\mathbf{z}$ and $\mathbf{A} \setminus \mathbf{H}$ certainly contains descendants of $\mathbf{z}$. It follows from Corollary 3.6 that $\hat{\mathbf{z}}_{\text{test}}$ will be a one-to-one mapping of $\mathbf{z}$ and can predict it perfectly at Step 14. Therefore, $\mathbf{z}$ will not be updated to $\mathbf{A}$ at this iteration.

For Case B.2-iii, we will show that the update takes place for $\mathbf{z}$ and its coparents. Without loss of generality, we assume that $\mathbf{z}$ only has one coparent $\mathbf{z}'$. As both $\mathbf{z}$ has multiple pure children (i.e., Condition 2.4-i) and all of them are in $\mathbf{A}$, the dictionary $\mathbf{P}$ will contain $\mathbf{z}$ and $\mathbf{z}'$ in their keys and $\mathbf{P}(\mathbf{z})$ and $\mathbf{P}(\mathbf{z}')$ contain their pure children respectively. Suppose $\mathbf{a}$ is a shared child of $\mathbf{z}$ and $\mathbf{z}'$, then $\mathbf{P}(\mathbf{a})$ would be a singleton set containing a one-to-one mapping of $\mathbf{z}'' := [\mathbf{z}, \mathbf{z}']$, after Step 6. At Step 9, $\mathbf{z}''$ will be recognized as a combination of $\mathbf{z}$ and $\mathbf{z}'$ and replaced with them. It follows that **JointP** will contain a pair $(\mathbf{Z}, \mathbf{H})$ such that $\mathbf{Z} = (\mathbf{z}, \mathbf{z}')$ and $\mathbf{H}$ is composed of all the children of the two coparents. At Step 16, both $\mathbf{z}$ and $\mathbf{z}'$ will be updated into $\mathbf{A}$. Cases with more than one coparents can be dealt with in the same manner.

For Case B.2-iv, we show that the update will not be carried out for $\mathbf{z}$ and its coparents. Following the notation in Case B.2-iii, we assume that $\mathbf{z}$ has a coparent $\mathbf{z}'$, and these two share at least one active child. If neither $\mathbf{z}$ nor $\mathbf{z}'$ has any pure active children, they will be regarded as a supernode $\mathbf{z}'' = [\mathbf{z}, \mathbf{z}']$ and the corresponding $\mathbf{H}$ contains only the shared active children. Then, $\mathbf{A} \setminus \mathbf{H}$ will certainly contain descendants of both $\mathbf{z}$ and $\mathbf{z}'$. Consequently, $\hat{\mathbf{z}}_{\text{test}}$ will be equivalent to the supernode $\mathbf{z}''$ and predicts it perfectly at Step 14. In this case, neither $\mathbf{z}$ and $\mathbf{z}'$ will be updated into $\mathbf{A}$. If only one of $\mathbf{z}$ and $\mathbf{z}'$ has no pure children in $\mathbf{A}$, then neither will be updated into $\mathbf{A}$ due to Step 9. Without loss of generality, we assume that $\mathbf{z}'$ does not have any active pure children, whereas $\mathbf{z}$ does. Then, there will be no estimated $\mathbf{z}'$ in the values of $\mathbf{P}$, but there will be estimated $\mathbf{z}$ and $(\mathbf{z}, \mathbf{z}')$. As a results, Step 9 will prevent $\mathbf{z}$ and $\mathbf{z}'$ from updated to $\mathbf{A}$. If both $\mathbf{z}$ and $\mathbf{z}'$ have active pure children, then there will be a pair $(\mathbf{Z}, \mathbf{H})$ in **JointP** such that $\mathbf{Z} = \{\mathbf{z}, \mathbf{z}'\}$ and $\mathbf{H}$ contains all their active children. As $\mathbf{H}$ does not contain all the children of $\mathbf{z}$ and $\mathbf{z}'$, the $\mathbf{A} \setminus \mathbf{H}$ will contain descendants of at least one of $\mathbf{z}$ and $\mathbf{z}'$. Then $\hat{\mathbf{z}}_{\text{test}}$ will contain all information of at least one of $\mathbf{z}$ and $\mathbf{z}'$ and predicts it perfectly at Step 16, which aborts the update of $\mathbf{z}$ and $\mathbf{z}'$. Therefore, no updates will happen in this case.

As each of the newly estimated variables in the values of **JointP** is a one-to-one mapping to a distinct variable in $\mathbf{Z}$, the iteration preserves Condition B.1-i.

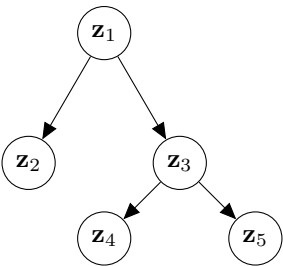

Figure 7: A specific case to demonstrate how we can avoid assuming specific functional classes.

For Condition B.1-ii, we can observe that if $\mathbf{z}$ is updated into $\mathbf{A}$ at this iteration, all its children would be in $\mathbf{A}$ at the beginning of this iteration and get removed at Step 16. Therefore, the update would not introduce directed paths to $\mathbf{A}$. Condition B.1-iii holds for the same reason: the newly introduced variables in the active set $\mathbf{A}$ at the end of the iteration separate their children in the active set $\mathbf{A}$ at the beginning of the iteration and all undiscovered latent variables.

So far, by induction, we have shown that Condition B.1-iii, Condition B.1-ii, and Condition B.1-iii hold at the end of each iteration. Given the analysis above, at each iteration, we update variables whenever all their children and their coparents' children are active, i.e., Case B.2-i and Case B.2-iii, and the causal relations between the parents and the children are encoded in **JointP**.

We further argue that at every iteration, there exists at least one undiscovered latent variable that has all its children in $\mathbf{A}$. We focus on the trimmed graph of $\mathbf{Z_A}$ and $\mathbf{A}$, where $\mathbf{Z_A}$ refers to all undiscovered variables. As the graph is of finite size and all variables in $\mathbf{Z_A}$ have directed paths ending at active variables in $\mathbf{A}$, the active variable $\mathbf{a}$ farthest from the root does not have siblings in $\mathbf{Z_A}$; otherwise, there would be $\mathbf{a}'$ (i.e., the child of $\mathbf{a}$'s sibling in $\mathbf{Z_A}$) in $\mathbf{A}$ that was further from the root than $\mathbf{a}$.

In sum, we have shown that there will exist at least one update at each iteration, and each update will lead to the discovery of new latent variables and correct causal structures. Therefore, Algorithm 1 can successfully identify the underlying hierarchical causal graph and each latent variable up to one-to-one mappings.

$\square$

## B.4 Illustration of a Specific Case

Figure 7 shows a specific case to demonstrate how we can avoid assuming specific functional classes. For active set $\mathbf{A} = \{\mathbf{z}_2, \mathbf{z}_4, \mathbf{z}_5\}$, if we treat $\mathbf{z}_2$ as $\mathbf{v}_1$, $\mathbf{z}_1$ will become $\mathbf{z}$ in the basis and $[\mathbf{z}_4, \mathbf{z}_5]$ will become $\mathbf{v}_2$. This set of $(\mathbf{z}, \mathbf{v}_1, \mathbf{v}_2)$ satisfies Assumption 2-i, as all information about $\mathbf{z}_1$ is contained by $\mathbf{z}_2$, $\mathbf{z}_4$, and $\mathbf{z}_5$. Assumption 2-iii and Assumption 2-ii are direct consequences of Assumption 3.3-iv and Assumption 3.3-iii. Also, we can always find an independent variable $\mathbf{s}_1 := \varepsilon_{z_2}$ in the basis model. Therefore, Theorem 3.4 guarantees the estimated $\hat{\mathbf{z}}$ to be a one-to-one mapping of $\mathbf{z}_1$.

# C Relaxation of Structural Conditions

## C.1 Relaxation of Condition 2.4-ii

We remark that Condition 2.4-ii is introduced to simplify the presentation of our main results – as the first step of nonlinear-latent-hierarchical causal discovery, we wish to present the basic techniques more clearly – handling triangles involves additional components of the algorithm and could potentially compromise the readability.

We give an example in Figure 8 with a triangle structure to illustrate how our theory and algorithm can handle triangles. We refer to variables in a triangle under the following convention: 1) root: the variable which is the cause of the other two variables (e.g., $\mathbf{z}_1$ in Figure 8), 2) sink: the variable which is the effect of the other two variables (e.g., $\mathbf{z}_3$ in Figure 8), and 3) bridge: the variable which is the effect of the root variable and the cause of the sink variables (e.g., $\mathbf{z}_2$ in Figure 8).

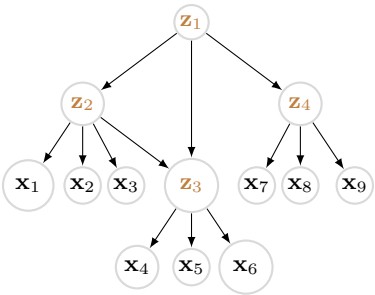

Figure 8: An example of triangle structures, i.e., $\mathbf{z}_1$, $\mathbf{z}_2$, and $\mathbf{z}_3$.

In the following, we illustrate the procedures to recognize and handle triangle structure by navigating through Figure 8. First, the directed path detection procedure portrayed in Section 3.3 can pinpoint the directed path between the bridge variable and the sink variable (e.g., $\mathbf{z}_2$ and $\mathbf{z}_3$). We need to determine whether the two variables indeed belong to a triangle. To this end, we can *lump* the bridge variable and the sink variable together as $\mathbf{v}_1$. We perform the basis model estimation over $\mathbf{v}_1$ and the active set $\mathbf{A}$ excluding $\mathbf{v}_1$'s children. In Figure 8, we perform the basis model over $\mathbf{v}_1 = [\mathbf{z}_2, \mathbf{z}_3]$ and $\mathbf{A} = \{\mathbf{z}_2, \mathbf{z}_3, \mathbf{x}_7, \mathbf{x}_8, \mathbf{x}_9\}$, which returns an estimate of $\mathbf{z}_1$. Further, we can determine that the bridge variable $\mathbf{z}_2$ and the estimated variable from the lumped $\mathbf{v}_1$ constitute the parents of the sink variable $\mathbf{z}_3$ by mutual prediction, which ascertains the two variables are indeed the bridge variable and the sink variable of a triangle, respectively. In this example, we find that the estimated variables of $\mathbf{z}_2$ and $\mathbf{z}_1$ together contain identical information to the estimated variable of $\mathbf{z}_3$, which reveals that $\mathbf{z}_2$ and $\mathbf{z}_3$ are the bridge variable and the sink variable of a triangle. This concludes the triangle detection procedure. With this knowledge, we can handle the triangle structure by lumping the bridge variable $\mathbf{z}_3$ and the sink variable $\mathbf{z}_3$ into a super-variable and proceed with this modification.

### C.2 Admitting Non-leaf variables into X

Identical to the directed path detection detailed in Section 3.3, Corollary 3.6 can detect non-leaf observed variables and exclude them from the initial active set $\mathbf{X}$ in Algorithm 1. Thus, the problem can be reduced back to the case without observed non-leaf variables, which can be tackled by Algorithm 1.

## D  Algorithm 1 Complexity

The dominant complexity term in our proposed algorithm is deep learning training, which is of a complexity $\mathcal{O}(R * n)$ where $R$ is the number of levels in the hierarchical model, and $n$ is the number of observed variables. In contrast, recent works on linear hierarchical models provide algorithms with complexity $\mathcal{O}(R * (1 + n)^{p+1})$ [Huang et al., 2022] and $\mathcal{O}(R * n!)$ [Xie et al., 2022] where $p$ is the largest number of variables that share children. Our algorithm achieves the lowest complexity in terms of the algorithmic complexity. As we allow for multi-dimensional variables, the dimensionality of each variable also contributes to the overall wall-clock time through the deep learning model dimension, which is a hyperparameter for the experimenter.

## E  Experiments

### E.1 Additional Experimental Details

We provide additional details of experiments in Section 4. For the basis model evaluation in Section 4.2, then encoder $\hat{f}$ and decoders $(\hat{g}_1, \hat{g}_2)$ are 4-layer MLP's with a hidden dimension 30 times as large as the input data and Leaky-ReLU ($\alpha = 0.2$) activation functions. For the synthetic hierarchy experiments in Section 4.3, the model layers are 8 and 50 times as large as the input data. For the personality dataset in Section 4.3, the model layers are 4 and 8 times as large as the input data. For the multi-view digit dataset in Section 4.3, the model layers are 4, and 4 times (encoder) and 2 times (decoder) as large as the input data The model configurations are summarized in Table 3.

| | # encoder layers | # decoder layers | encoder widths | decoder width |
|---|---|---|---|---|
| Basis models (Section 4.2) | 4 | 4 | 30× | 30× |
| Synthetic hierarchy (Section 4.3) | 8 | 8 | 50× | 50× |
| Personality dataset (Section 4.3) | 4 | 4 | 8× | 8× |
| Digit dataset (Section 4.3) | 4 | 4 | 4× | 2× |

Table 3: **Estimation model configuration for each experiment.**

We apply Adam to train each model for $20,000$ steps with a learning rate of $1e-3$. For hierarchical models, we consistently use 2-layer MLPs for generating functions and set the exogenous variable dimensionality as 2. We evaluate the $R^2$ score with kernel regression over $8192$ samples for each estimated variable pair. Figure 6.b, The matched pairs give significantly higher scores than the unmatched ones (e.g., Table 2 and Figure 6b). For instance, in the first row of Table 2, the estimated latent variable with This gap allows us to set a threshold for each experiment to distinguish the matched estimated variables (e.g., a threshold of $0.6$ suffices for the experiment in Table 2).

### E.2 Additional Results on the Basis Model

Figure 9 visualizes the dependence between the estimated and true components. We can observe that the $\mathbf{z}$ is highly correlated with $\hat{\mathbf{z}}$ whereas $\mathbf{s}_1$ has little correlation with $\hat{\mathbf{z}}$, which verifies Theorem 3.2 that we can successfully identify $\mathbf{z}$.

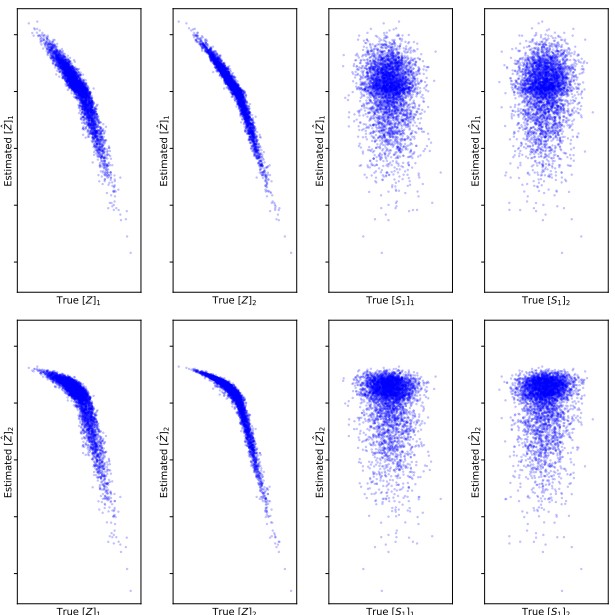

Figure 9: **The scatter plot between the estimated $\hat{\mathbf{z}}$ and the true $(\mathbf{z}, \mathbf{s}_1)$.** The scatter points are obtained in an estimation run with $d_z = d_{s_1} = d_{s_2} = 2$. We can observe that the estimated components of $\hat{\mathbf{z}}$ are highly correlated with those of $\mathbf{z}$ and have little correlation with those $\mathbf{s}_1$, empirically verifying Theorem 3.2.

Although our theory only concerns asymptotic properties, our approach can perform stably at relatively low-sample regimes. As shown in Figure 10, our algorithm can achieve decent identification scores when the sample number exceeds $5000$. The performance peaks with little variance when the sample size is over $10000$.

### E.3 Additional Results on Synthetic Hierarchical Models.

Table 4 and Table 5 contain the mutual prediction scores for the hierarchical structures in Figure 4a and Figure 4c. We can observe that the estimated variables that yield high mutual prediction scores are indeed under the same causal cluster, which gives signals for structure learning.

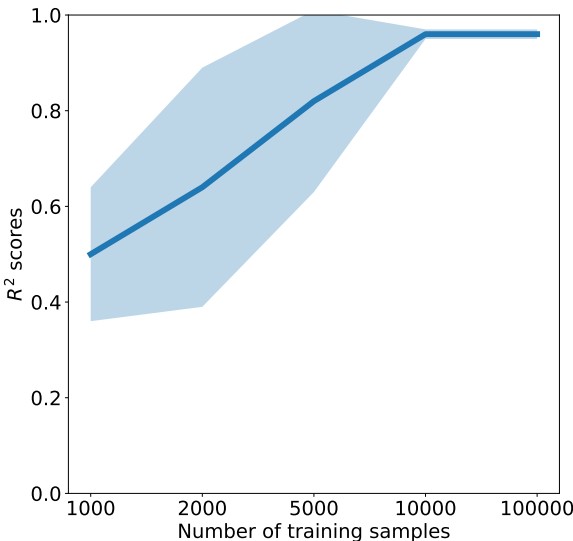

Figure 10: $R^2$ **scores for basis model identification.** The true basis model (Figure 2 in the manuscript) has latent dimensionality $d_z = d_{s_1} = d_{s_2} = 2$. The basis model training and data generation follow the training configuration in the main text. We vary the number of training samples and measure the $R^2$ score to assess the identification result. Each result is averaged over 3 random seeds.

|  | $\mathbf{x}_1$ | $\mathbf{x}_2$ | $\mathbf{x}_3$ | $\mathbf{x}_4$ |
|---|---|---|---|---|
| $\mathbf{x}_1$ | $\times$ | $\mathbf{0.83 \pm 0.03}$ | $0.57 \pm 0.04$ | $0.50 \pm 0.08$ |
| $\mathbf{x}_2$ | $\mathbf{0.86 \pm 0.01}$ | $\times$ | $0.58 \pm 0.03$ | $0.52 \pm 0.04$ |
| $\mathbf{x}_3$ | $0.6 \pm 0.02$ | $0.59 \pm 0.05$ | $\times$ | $\mathbf{0.77 \pm 0.00}$ |
| $\mathbf{x}_4$ | $0.59 \pm 0.02$ | $0.59 \pm 0.05$ | $\mathbf{0.81 \pm 0.04}$ | $\times$ |

Table 4: **Pairwise predictions among estimates in Figure 4a**. Each box $(\mathbf{a}, \mathbf{b})$ shows the $R^2$ score obtained by applying the estimate produced by treating $\mathbf{a}$ as $\mathbf{v}_1$ to predict the estimate produced by treating $\mathbf{b}$ as $\mathbf{v}_1$. We can observe that the prediction scores within sibling pairs $(\mathbf{x}_1, \mathbf{x}_2)$ and $(\mathbf{x}_3, \mathbf{x}_4)$ are noticeably higher than other pairs, showing a decent structure estimation.

### E.4 Additional Results on the Personality Dataset

We randomly and evenly partition each set of six questions into two variables. Following this procedure, we have observed ten variables $\mathbf{c}_1$, $\mathbf{c}_2$, $\mathbf{a}_1$, $\mathbf{a}_2$, $\mathbf{n}_1$, $\mathbf{n}_2$, $\mathbf{e}_1$, $\mathbf{e}_2$, $\mathbf{o}_1$, and $\mathbf{o}_2$, each of 3-dimension. The Letter in each variable name indicates the attribute to which the variable belongs, to facilitate readability. Figure 11 contains the results. Analogous to Figure 5, the learned structure faithfully reflects the questionnaire structure – questions designed for the same personality trait are clustered together.

|  | $\mathbf{z}_4$ | $\mathbf{x}_3$ | $\mathbf{x}_4$ | $\mathbf{x}_5$ |
|---|---|---|---|---|
| $\mathbf{z}_4$ | $\times$ | $\mathbf{0.81 \pm 0.011}$ | $0.48 \pm 0.006$ | $0.52 \pm 0.000$ |
| $\mathbf{x}_3$ | $\mathbf{0.87 \pm 0.002}$ | $\times$ | $0.47 \pm 0.003$ | $0.54 \pm 0.002$ |
| $\mathbf{x}_4$ | $0.57 \pm 0.000$ | $0.52 \pm 0.006$ | $\times$ | $\mathbf{0.77 \pm 0.005}$ |
| $\mathbf{x}_5$ | $0.58 \pm 0.001$ | $0.57 \pm 0.015$ | $\mathbf{0.81 \pm 0.000}$ | $\times$ |

Table 5: **Pairwise predictions among estimated variables in Figure 4c**. The pattern is consistent with Table 4 and Table 2.

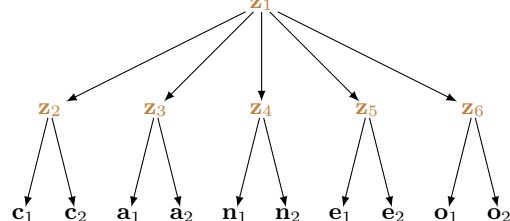

Figure 11: The casual graph learned by our method, where each observed variable ($\mathbf{c}$, $\mathbf{a}$, $\mathbf{n}$, $\mathbf{e}$, $\mathbf{o}$) is of three dimensions.

