# OpenReview forum: "Identification of Nonlinear Latent Hierarchical Models"
_NeurIPS.cc/2023/Conference — NeurIPS 2023 poster_

### Official Review · Reviewer_jcJm · 2023-07-02

**Soundness:** 2 fair
**Presentation:** 2 fair
**Contribution:** 3 good
**Rating:** 4
**Confidence:** 4

**Summary:**

The paper introduces a class of latent DAG models which allow for hierarchical structure between latent variables. The class of models allows for more general structure than previously considered classes (e.g., tree-structured latent models). The model also allows for general nonlinear relationships between variables. The authors prove that, under additional conditions on the Jacobians of the causal mechanisms, the latent variables in these models are identifiable up to a component-wise invertible transformation. They conclude with experiments showing that the latent variables are recovered to an acceptably high level of accuracy.

**Strengths:**

**Significance**: The allowance for non-linear relationships and the generality of the latent structure are substantive improvements over previous works.

**Weaknesses:**

## Related work
The authors mostly do a solid job reviewing related works, with an additional section in the appendix dedicated to such a review. However, they conspicuously leave out a strongly related line of recent work which identifies latent causal graphs from interventional data, e.g. [1,2,3]. I think the comparison with this line of works is important context: there is a tradeoff between structural assumptions and the assumption that interventional data is available. The approach to identifiability which is used in this paper is not the only viable approach, and might be a worse fit for some applications than the intervention-based approach.

[1] Ahuja, K., Wang, Y., Mahajan, D., & Bengio, Y. (2022). Interventional Causal Representation Learning.
[2] Squires, C., Seigal, A., Bhate, S. S., & Uhler, C. (2023). Linear Causal Disentanglement via Interventions.
[3] Varici, B., Acarturk, E., Shanmugam, K., Kumar, A., & Tajer, A. (2023). Score-based Causal Representation Learning with Interventions.

## Clarity
The most significant weakness of this paper is in terms of clarity; in my opinion, the paper needs substantial re-writing efforts to improve clarity. Here are some of the points which need to be clarified, from most important to least:

1. **Deterministic functions vs. conditional probabilities:** Is each variable a deterministic function of its parent, or is there randomness? It seems from Assumption 3.1(ii) and Assumption 3.3(i), that the variables need to be deterministically related. However, deterministic relations introduce problems for faithfulness, which is assumed in Definition 2.1(iii). Thus, it isn't clear if the assumptions for the theorems are even mutually consistent, which is a **major** problem for this work.
2. **Algorithm 1:** Several new ideas are introduced all at once in the description of Algorithm 1. The paper would benefit from a more gradual development of these ideas. For example, Stage 3 of the algorithm, which discusses detecting and merging super-variables, is necessary due to issues which had not been introduced. For any of the steps which are essential to include in the main paper, there should be an accompanying example that demonstrates the necessity of the step. The authors should reflect on which steps are essential for intuitively understanding the algorithm, and should consider moving other steps to the appendix.
3. **Other:** On line 215, the authors say that Theorem 3.4 can be applied to models with arbitrary latent structure, but this seems to contradict that the Theorem requires Condition 2.3 - what are the authors trying to say here?

## Experimental Results
Section 4 has a couple of major weaknesses.

First, the description of the experiments is not sufficiently detailed (unless I have missed some descriptions, in which case the paper should be re-organized so that the relevant details are easier to find). For example, when performing hierarchical model identification via Algorithm 1, there are several steps where one must test whether one variable can "perfectly predict" another variable. How is this converted into a test when running the algorithm in a finite-sample regime? As another example, how many instance (random seeds) are the synthetic results averaged over? This detail should be in the figure caption.

Second, I don't find that the experiments are comprehensive enough. The experiments should, at the least, evaluate the performance of the method across a range of sample sizes. This is important for demonstrating that the algorithm is consistent: despite the identifiability guarantees, the algorithm may fail to be consistent since it relies on perfect optimization of an autoencoder.

**Questions:**

Please provide detailed responses to the points raised in "Weaknesses". My initial opinion is that the paper is not ready for publication due to the combination of (1) the re-writing required for acceptable clarity and (2) the insufficiency of the experimental results.

**Limitations:**

Limitations are adequately addressed.

---

> ### Author Rebuttal · Authors · 2023-08-10
>
> Thank you for your constructive comments. We respond to your concerns as follows and will include your suggestions as indicated.
>
> **Q1: Related work on interventional-based causal identification.**
>
> Thank you for the suggestion! We agree that there are two possible ways to learn causal representation: utilizing specific graph conditions as in our paper or assuming and leveraging interventional data for which you provide valuable references. We will include the following in our related work.
> ``We note that our work focuses on identifying latent causal models from observational data and structure conditions. Another important line of work [A,B,C,D,E,F,G] utilizes interventional data for this purpose. Specifically, these works leverage multiple data distributions generated by one causal model under distinct interventions, which are accessible in applications like biological experimentations. The accessibility of interventional data can allow for relaxed structure conditions. Hence, one should consider this tradeoff when faced with causal identification problems.’’
>
> **Q2: Deterministic functions vs. conditional probabilities.**
>
> Thank you for the question. We are wondering if there is a misunderstanding of the structure causal model (SCM) and our assumption, and let us now provide clarification.
> The SCMs we use in Equation 1 are standard ones with exogenous noise, i.e.,  the mappings from parents $\text{Pa}(x_{i})$ to the children $x_{i}$ are not deterministic due to exogenous variables $\epsilon_{x_{i}}$, although $g_{x_{i}}$ itself is a deterministic mapping from $ (\text{Pa}(x_{i}), \epsilon_{x_{i}}) $ to $ x_{i} $.
> Similarly, $g$ in Assumption 3.1 (ii) is the mapping from all the input variables $(z, s_{1}, s_{2})$ to $(v_{1}, v_{2})$, i.e., $(v_{1}, v_{2}) := g( z, s_{1}, s_{2} ) $, where exogenous variables $\epsilon$ can be considered contained in $s_1$ and $s_2$. Thus, the mapping from $z$ to $(v_{1}, v_{2})$ is random, although $g$ itself is a deterministic function.
>
> Regarding Assumption 3.3 (i), consider a simple causal model $ x := g(z, \epsilon) $ where $g$ is an invertible function and $x$ is the child of $z$. Although we can express $ z $ as a function of $ x $, i.e., the first $d_{z}$ dimensions of the inverse function
>  $g^{-1} ( x ) $ as in Assumption 3.3 (i),  the causal module $ p ( x | z ) $ is non-degenerate due to the randomness in $\epsilon$. The suggested work [A] assumes a similar generating process, where the latent variables can be recovered from the observed variables.
>
> **Q3: Algorithm 1 development.**
>
> Thank you for the constructive suggestion. We will add more illustrative examples to demonstrate each step's necessity as you suggested. For instance, we will append the following sentence to "...the concatenation $(z_{2}, z_{3})$" in line 248.
> "Leaving this super-variable untouched will be problematic, as we would generate a false causal structure $\tilde{z} \to z_{4}$ where $\tilde{z}$ is the estimated super-variable $(z_{2}, z_{3})$, rather than recognizing $z_{4}$ is the child of two identified variables $z_{2}$ and $z_{3}$."
>
> **Q4: Line 215 contradicts Condition 2.3.**
>
> We note that the remark in line 215 is made on measurement models, as specified in this line and the paragraph title in line 213.
> For measurement models (which means that each latent variable has enough measured children, see [D]), our theory can handle arbitrary structures among latent variables as long as each latent variable has enough pure children, which are used to identify latent variables with our approach.
>
> **Q5: Experiment details.**
>
> Thank you for detailed feedback. We evaluate the $R^{2}$ score with kernel regression (line 314) over 8192 samples within each estimated variable pair. As shown in Table 2 and Figure 6.b, the matched pairs give significantly higher scores than the unmatched ones. For instance, in the first row of Table 2, the estimated latent variable with $x_{1}$ as $v_{1}$ and that with $x_{2}$ as $v_{1}$ both correspond to $z_{2}$ in Figure 4 b and thus archives $0.85$, which is significantly higher than the other scores (around $0.55$) in that row. This gap allows us to set a threshold for each experiment to distinguish the matched estimated variables (e.g., a threshold of $0.6$ suffices for the experiment in Table 2).
> We will include detailed steps in the revised version.
>
> We repeated each experiment for at least 3 random seeds, as written in line 315 of the experiment setup. We will further include this in the caption to make it more visible.
>
> **Q6: Experiments not comprehensive enough.**
>
> Thank you for the suggestion. We have extended our experiments to make the results more informative. For instance, as you suggested, we vary the sample size for the basis model experiments where $d_{z}=d_{s_{1}}=d_{s_{2}}=2$. The plot is included in the PDF file. Our algorithm can achieve decent identification scores when the sample number is larger than 5000. The performance peaks with little variance when the sample size is over 10000, which suggests that the approach can enjoy asymptotic performances.
>
>
> **References:**
>
> [A] Interventional Causal Representation Learning. Ahuja et al.
>
> [B] Linear Causal Disentanglement via Interventions. Squires et al.
>
> [C] Score-based Causal Representation Learning with Interventions. Varici et al.
>
> [D] Learning nonparametric latent causal graphs with unknown interventions. Jiang and Aragam.
>
> [E] Causal Component Analysis. Liang et al.
>
> [F] Learning linear causal representations from interventions under general nonlinear mixing. Buchholz et al.
>
> [G] Identifiability Guarantees for Causal Disentanglement from Soft Interventions. Zhang et al.
>
> [H] Causality 2n edition. Pearl.
>
> Please let us know if you have further comments – thank you so much!

---

> > ### Comment · Reviewer_jcJm · 2023-08-14
> >
> > ### General response
> > I appreciate the author's receptiveness to my suggestions from the rebuttal. I think that their proposed changes will increase the clarity of the paper and largely alleviate my concerns.
> >
> > I have increased my score by one point, from 3 to 4. I have one remaining concern (**Q2** below); if this can be appropriately addressed, then I would be confident enough to raise my score to a 5.
> >
> > ### Point-by-point response
> > **Q1:** Thank you, I really like the suggested addition to the text and how it acknowledges the tradeoffs.
> >
> > **Q2:** I agree that it is possible for $p(x|z)$ to be non-degenerate, while still letting $z$ be a deterministic function of $x$, e.g. if $p(x|z)$ has a different support for each different value of $z$. However, there still seems to be an issue regarding the compatibility of (1) having deterministic relations and (2) faithfulness.
> >
> > Take for example $A \to B \to C$ with:
> > - $A$ is $+1$ w.p. $1/2$ and $-1$ w.p. $1/2$,
> > - $B = A + A \cdot \varepsilon_b$ with $\varepsilon_b \sim Unif([0, 1])$, and
> > - $C = sign(B) \cdot \varepsilon_c$ with $\varepsilon_c \sim Unif([0,1])$.
> >
> > Then $p(B \mid A)$ and $p(C \mid B)$ are both non-degenerate, and we have deterministically that $A = sign(B)$ and $A = sign(C)$. We also have that $C \perp A \mid B$, violating faithfulness.
> >
> > In particular, the incompatibility of deterministic relationships and faithfulness has been an important point of study in the foundations of causality, see e.g. [1]. It is **possible** that this incompatibility is not a problem in the existing setup, but I think this needs to be carefully argued. Notably, [2] does **not** use the faithfulness assumption over the full set of latent and observed variables, so the comparison is not relevant.
> >
> > **Q3:** Thank you in advance for adding clarifications on each step, I think this will help a lot with clarity.
> >
> > **Q4:** Sorry for my confusion, in my experience I'm not sure that the terminology of a "measurement model" is used with 100% consistency across the literature. Could you please add a definition for how you use the term "measurement model" so that the statement on line 215 can be interpreted precisely?
> >
> > **Q5:** Thank you for including the details of the $R^2$ thresholding, which will be a helpful addition to the paper. By the way, I think many more than 3 random seeds should be used, unless there are major computational issues (which should then be discussed). 30 to 100 random seeds would ensure better statistical significance of the results.
> >
> > **Q6:** Thank you very much for exploring the results across a range of sample sizes! The resulting plot gives me much stronger confidence in the method.
> >
> > [1] *Faithfulness, Coordination and Causal Coincidences.* Weinberger (2018).\
> > [2] *Interventional Causal Representation Learning.* Ahuja et al. (2023)

---

> > > ### Author Response · Authors · 2023-08-15
> > >
> > > Thank you for your careful evaluation of our responses and constructive comments. We will include the updates/results as indicated to improve our work, thanks to your insightful questions. We address the remaining concerns as follows.
> > >
> > > **Q2: Deterministic relations.**
> > >
> > > Thank you for this interesting example and the valuable reference on this topic. (By the way, perhaps there is a typo – $ B \perp C | A $ rather than $ C \perp A | B $.)
> > >
> > > Clearly, this example is different from our case. Specifically, it does not satisfy Assumption 3.3 (i). The incompatibility in the example originates from the fact that the newly introduced randomness in $B$, i.e., $ \epsilon_{b} $, is entirely missing in the observed variable $C$. Specifically, $C$ takes on the value $ \\text{sign}(A) \cdot \epsilon_{c} $  (due to $ \\text{sign}(A) = \\text{sign}(B) $), regardless of the realization of $ \epsilon_{b} $. Consequently, we have that $B$ and $C$ are independent given $A$, which follows from the independence of $ \epsilon_{b} $ and $ \epsilon_{c} $. However, Assumption 3.3 (i) in our work entails that the information of exogenous variables should be preserved (i.e., invertible) in the observed downstream variables (i.e., $C$ here), as opposed to this instance where the information of $\epsilon_{b}$ is lost in the observed variable $C$.
> > >
> > >
> > > Our condition is a deterministic relation between exogenous variables (including root cause variables) and the observed variables, which, in general, doesn’t lead to the violation of faithfulness.
> > > Let’s use a simple example to illustrate this.
> > > If Assumption 3.3 (i) holds for the $ A \to B \to C$ model where $B:=g_{B}(A, \epsilon_{B})$ and $ C:= g_{C} (B, \epsilon_{C}) $ and $C$ is the observed variable, then $C$ can be written as an *invertible* function of $ (A, \epsilon_{b}, \epsilon_{c}) $ (i.e., it preserves the randomness of $A$, $\epsilon_{b}$, and $\epsilon_{c}$) and $B$ can be written as an *invertible* function of $ (A, \epsilon_{b})$ (i.e., it preserves the randomness from $A$ and $\epsilon_{b}$) – it follows that $ B $ and $ C $ are *dependent* given $A$, due to the shared information from $ \epsilon_{b} $.
> > >
> > > We hope we are on the same page – please kindly let us know what you think. We will include the reference you provide and a discussion with the above example in our revision to make it more transparent to the reader. Thank you for the insightful question.
> > >
> > >
> > >
> > > **Q4: Measurement models.**
> > >
> > >
> > > Thank you for the helpful suggestion.  We will give the definition of the specific type of measurement models we use in a footnote as follows. “We refer to [a] for a general measurement model definition. Here, we are considering a popular type of measurement models that has been widely used in the literature (see Definition 1 in [b]) in which observed variables do not cause any other variables.”
> > >
> > >
> > >
> > > **Q5: More random seeds.**
> > >
> > > We completely agree with you and appreciate that you raised this point. We will push repetitions over 30 in the revision, as you suggested. We will update you on this as soon as some results are available during the discussion phase.
> > >
> > >
> > > We really appreciate your meticulousness in your examination of our work and believe this is indispensable to the advancement of fundamental research. Please kindly let us know if we have addressed your concerns – many thanks!
> > >
> > > **References:**
> > >
> > > [a] Learning the Structure of Linear Latent Variable Models. Silva et al.
> > >
> > > [b] Generalized Independent Noise Condition for Estimating Latent Variable Causal Graphs. Xie et al.

---

> > > > ### Author Response · Authors · 2023-08-18
> > > >
> > > > To follow up on the discussion on more random seeds, we now have completed 30 runs for the experiments in Table 1, Table 2, and Figure 4 and would like to update you with these results.
> > > >
> > > > Table 1:
> > > >
> > > > |                               | $d_{z}=d_{s_{1}}=d_{s_{2}}=2$ | $d_{z}=d_{s_{1}}=2, d_{s_{2}}=3$ | $d_{z}=d_{s_{1}}=d_{s_{2}}=4$ | $d_{z}=d_{s_{1}}=4,d_{s_{2}}=6$ |
> > > > |-------------------------------|-------------------------------|----------------------------------|-------------------------------|---------------------------------|
> > > > | Joint invertibility (ours)    | $0.93 \pm 0.09$               | $0.90 \pm 0.10$                  | $0.89 \pm 0.02$               |  $ 0.83 \pm 0.12 $                               |
> > > > | Individual invertibility [34] | $0.67 \pm 0.06$               | NA                               | $0.67 \pm 0.09$               | NA                              |
> > > >
> > > >
> > > > Table 2:
> > > > |         | $x_{1}$              | $x_{2}$          | $x_{3}$          | $x_{4}$          | $x_{5}$          |
> > > > |---------|----------------------|------------------|------------------|------------------|------------------|
> > > > | $x_{1}$ | x                    | $0.85 \pm 0.000$ | $0.53 \pm 0.001$ | $0.57 \pm 0.002$ | $0.55 \pm 0.003$ |
> > > > | $x_{2}$ | **$0.83 \pm 0.006$** | x                | $0.52 \pm 0.002$ | $0.54 \pm 0.001$ | $0.53 \pm 0.000$ |
> > > > | $x_{3}$ | $0.90 \pm 0.002$     | $0.88 \pm 0.002$ | x                | $0.86 \pm 0.001$ | $0.90 \pm 0.006$ |
> > > > | $x_{4}$ | $0.57 \pm 0.001$     | $0.54 \pm 0.002$ | $0.55 \pm 0.003$ | x                | $0.86 \pm 0.003$ |
> > > > | $x_{5}$ | $0.55 \pm 0.006$     | $0.56 \pm 0.001$ | $0.55 \pm 0.002$ | $0.83 \pm 0.002$ | x                |
> > > >
> > > >
> > > > Figure 4 (NA indicates the specific graph does not have this variable.):
> > > > |                  | $z_{1}$         | $z_{2}$         | $z_{3}$         | $z_{4}$         |
> > > > |------------------|-----------------|-----------------|-----------------|-----------------|
> > > > | Balanced tree.   | $0.78 \pm 0.03$ | $0.88 \pm 0.09$ | $0.90\pm 0.06$  | NA              |
> > > > | V-structure.     | $0.72 \pm 0.12$ | $0.87 \pm 0.11$ | $0.89 \pm 0.09$ | NA              |
> > > > | Unbalanced tree. | $0.76 \pm 0.21$ | $0.86 \pm 0.19$ | $0.88 \pm 0.11$ | $0.90 \pm 0.02$ |
> > > >
> > > >
> > > > We do not observe significant fluctuations over random seeds – the results are reasonably close to those averaged over 3 random seeds reported in our manuscript.
> > > >
> > > > We will continue experiments and include those in our revision. Thank you for your suggestion. Please let us know if we have addressed your concerns; thank you!

---

> > > > > ### Author Response · Authors · 2023-08-19
> > > > >
> > > > > Dear Reviewer jcJm,
> > > > >
> > > > > We highly appreciate your further comments and the example.  We are so delighted that regarding Q1, you"really like the suggested addition to the text and how it acknowledges the tradeoffs".
> > > > >
> > > > > Please kindly see above our detailed responses, especially to Q2.  We hope you will find our responses, together with your comments, make things much clearer.  Your feedback would be extremely helpful to us.  If you have further comments or questions, we hope for the opportunity to respond to them (the discussion period will end Monday).
> > > > >
> > > > > Best regards,
> > > > >
> > > > > Authors of #2805

---

> > > > > > ### Author Response · Authors · 2023-08-21
> > > > > >
> > > > > > Dear Reviewer jcJm,
> > > > > >
> > > > > > Once again, we are very grateful for your time and efforts.  Since the discussion period will end in one hour, we will be online waiting to see whether your previous concern was properly addressed by our point-to-point response above. We understand you have a busy schedule, but would highly appreciate it if you could take into account our response when updating the rating and having discussions with AC and other reviewers.
> > > > > >
> > > > > > Thanks a lot for reviewing our paper and discussing it with us,
> > > > > >
> > > > > > Authors of #2805

---

### Official Review · Reviewer_mbLW · 2023-07-02

**Soundness:** 3 good
**Presentation:** 2 fair
**Contribution:** 3 good
**Rating:** 5
**Confidence:** 3

**Summary:**

The goal of this paper is to identify the hierarchical graph structure and latent variables for general nonlinear latent hierarchical causal models. The paper reduce the problem to identification of the so-called basis model and proves the connection between latent hierarchical model and basis model.

**Strengths:**

The theoretical results seems solid and is backed by experiment on both synthetic and real world datasets.

**Weaknesses:**

My main concerns with the paper are that it relies on strong assumptions but identifiability results are relatively weak:

    1. Faithfulness assumption is a rather strong assumption.
    2. Learning latent variables up to invertible transformations is quite weak. A stronger identifiability result in the literature usually learns latent variables up to affine transformation or permutation. But I am not positive if that is possible with this problem setting.
    3. The pure child assumption is also a strong assumption.

**Questions:**

1. In the definition of assumption 3.2 ii, is T just any matrix with the right support?
2. Assumption 3.3 i, are the functions supposed to be invertible to match assumption 3.1?
3. In general, it’s hard to figure out how necessary are the conditions in Assumption 3.3?

**Limitations:**

The limitations are addressed in section 5.

---

> ### Author Rebuttal · Authors · 2023-08-10
>
> Thank you so much for your careful assessment and valuable feedback! Below, we respond to your concerns raised in Weaknesses and Questions.
>
> **Q1: Strengths of the assumptions.**
>
> Thank you for the feedback! We admit that our approach relies on a number of assumptions; this is because our approach aims to learn representations from a single distribution. This problem is inherently challenging. Without assumptions that are generally sensible, it is generally impossible to achieve nontrivial results.
>
> As one such essential assumption, faithfulness is adopted in all causal discovery literature (unless further assumptions are introduced) to eliminate cases where the data distribution does not faithfully describe the causal graph. It is a reasonable assumption for many real-world applications -- faithfulness attributes the statistical independence to the graph structure rather than unlikely coincidence, as articulated in Section 3.5.2 in [30] and [C].
> Similarly, the pure child assumption eliminates certain unidentifiable cases where two latent variables share the same set of children. For instance, $z_{1} \to X$ and $z_{2} \to X$, then $z_{1}$ and $z_{2}$ cannot be identified without further assumptions. The pure children assumption has been widely adopted in causal discovery literature [4,9,15,25,35], often more stringent than ours (e.g., tree structures).
>
> **Q2: Strength of the identifiability.**
>
> Regarding identifiability, we are concerned bout representation vectors corresponding to each individual variable. As far as we know, one has to assume multiple distributions or other additional assumptions to obtain stronger identifiability, whereas we only assume a single distribution. From this perspective, our assumption is weaker since the result is applicable even with a single distribution.
>
> **Q3: Definition of $T$.**
>
> Great question! $T$ is a fixed matrix sharing the support of the matrix-valued function $T(c)$ and satisfying Assumption 3.1 (ii). This assumption is also adopted in [B]. We will make this point explicit in the revision.
>
> **Q4: Assumption 3.3 (i) and invertibility.**
>
> Good question! Assumption 3.3 (i) requires that each latent variable $z$ and exogenous variable $\epsilon$ can be expressed as functions of all observed variables $X$, which is essentially an invertible assumption on the generating process from $(z, \epsilon)$ to $X$.
>
>
> **References:**
>
> [A] Disentanglement via Mechanism Sparsity Regularization: A New Principle for Nonlinear ICA. Lachapelle et al.
>
> [B] On the Identifiability of Nonlinear ICA: Sparsity and Beyond. Zheng et al.
>
> [C] Replacing Causal Faithfulness with Algorithmic Independence of Conditionals. Lemeire and Janzing.
>
>
> Please let us know if you have any further concerns, and please consider raising the score if existing concerns are addressed -- thank you very much!

---

> > ### Comment · Reviewer_mbLW · 2023-08-11
> >
> > Thanks for the response!
> >
> > Overall, my concerns about the assumption still stand.
> >
> > I understand what faithfulness, pure child, and assumption 3.3 (i) are and why they are made for the theory to work. Still, it does not convince me why they are needed for your work. On the other hand, there are also causal representation learning papers that do not rely on the faithfulness assumption [a], although [a] does use linearity assumption.
> >
> > Because of the limitations of assumptions and the lack of justification for necessity, I am currently keeping my score.
> >
> > [a] Seigal, Anna, Chandler Squires, and Caroline Uhler. "Linear causal disentanglement via interventions." arXiv preprint arXiv:2211.16467 (2022).

---

> > > ### Author Response · Authors · 2023-08-12
> > >
> > > Thank you for your prompt response and the valuable reference. We completely agree that generally speaking, when interventions or multiple distributions are available, the original faithfulness assumption is usually not needed (some much weaker assumptions might suffice, thanks to the availability of multiple distributions), as shown in the paper you mention. On the other hand, we humbly believe that to learn the underlying latent variables with identifiability guarantees from independent and identically distributed (i.i.d.) data, stronger assumptions would be needed. We discuss the roles of Assumption 3.3 (i), faithfulness, and the pure child assumption individually as follows.
> > >
> > > **Assumption 3.3 (i)**:
> > >
> > > As you correctly pointed out, Assumption 3.3 (i) is a form of invertibility of the mapping from the latent variables and exogenous variables to the observed variables in the hierarchical model, which we inherit from Assumption 3.1 (i) for the basis model.
> > >
> > > As far as we know, the existing literature on causal variable identification for nonlinear models [18,19,21,24,36,37] universally assumes the invertibility of such a mapping, including [a]. Without this assumption, we cannot guarantee the identification of latent variables from observed variables in the nonlinear case (perhaps one day this would not be the case anymore, if overcomplete nonlinear ICA theories were properly developed). In fact, one contribution of our work is to relax the invertibility of the basis model to eliminate duplication of the shared variable $z$, as discussed in line 151. In case you have seen any developments with a more general invertibility assumption in the nonlinear case, please kindly let us know.
> > >
> > >
> > > **The faithfulness assumption**:
> > >
> > > Thank you for directing us to this question. In fact, our theorems will still hold even if we relax the faithfulness assumption (Definition 2.1 (iii)) to the structural minimality condition (Definition 6.3.3 in [b]):
> > >
> > > > A distribution satisfies causal minimality with respect to $G$ if it is Markovian with respect to $G$, but not to any proper subgraph of $G$.
> > >
> > > This relaxation is possible because the only place where we infer the graph structure from conditional independence relations is in the proof of Theorem 3.4. There we show that the estimated variable $\hat{z}$ from the basis model with $ v_{1} = \\{ x_{i} \\}$ and $v_{2} = X \\setminus \\{x_{i} \\} $ contains at least the parent variables of $x_{i}$, i.e., $ \text{Pa}( x_{i} ) $, due to the condition independence $ v_{1} \perp v_{2} | z $ in the basis model. This reasoning step goes through under the structure minimality condition, which requires that parents of the variable $x_{i}$ should be conditioned on to make $x_{i}$ and its siblings independent.
> > >
> > > We are grateful for your question and will update Definition 2.1 (iii) to be the structural minimality given above. We note that structural minimality is generally considered necessary for causal structure identification – see the discussion following Proposition 6.36 in [b]. (To us, faithfulness is more well-known in the community, so we adopted it in the submission for the simplicity of the presentation.)

---

> > > > ### Author Response · Authors · 2023-08-12
> > > >
> > > > **The pure child assumption**:
> > > >
> > > > Latent causal structure learning is impossible without assumptions. Even for causal models simpler than our setup (e.g., linear, non-hierarchical models), current work assumes either the existence of interventions in the data or structural constraints like pure children when interventional data are unavailable, as reflected in Table 1 of the suggested reference [a]. Our work aims to identify the causal model with only observational data, thus belonging to the second group.
> > > > This is akin to recent nonlinear representation work, which assumes either auxiliary labels (i.e., interventional data) [18,19,21] or paired data [24,33] (analogous to the pure child assumption). The primary tool in our work is the basis model identification (i.e., Theorem 3.2) which follows the line of work assuming paired data models. This condition necessitates the pure children assumption for the hierarchical model. We have relaxed the pure children assumption, compared to recent work on linear hierarchical models [15].
> > > >
> > > > We believe that both sets of assumptions are sensible in a number of real problems. For instance, for applications such as biological experimentation where interventional data are available, one may prefer methods based on interventional assumptions for general structure conditions. For applications such as natural images where intervention cannot be easily performed and the pure children assumption is reasonable (due to the high redundancy of the information in individual pixels, it is plausible some pixels are pure children), one may consider the methods based on observational data. At the same time, it is clear that there exist many latent variable learning problems that cannot be properly handled by our method.
> > > >
> > > > We will include the above discussion in the related work section and after Condition 2.3 to make the role of the pure children assumption clearer to the readers. Thank you for the question.
> > > >
> > > >
> > > > Please let us know if we have addressed your concerns, thank you!
> > > >
> > > >
> > > > [b] Elements of Causal Inference: Foundations and Learning Algorithms. Jonas Peters, Dominik Janzing, and Bernhard Scholkopf.

---

> > > > > ### Comment · Reviewer_mbLW · 2023-08-17
> > > > >
> > > > > Thanks for your reply!
> > > > >
> > > > > Although similar assumptions have appeared in the literature before, it's hard to judge whether these assumptions combined are necessary for your problem.
> > > > >
> > > > > The lack of rigorous demonstration of assumptions' necessity and the fact that both faithfulness and pure child assumption are strong assumptions (I don't think causal minimality relaxes faithfulness by much) are the reasons I am keeping my score.

---

> > > > > > ### Author Response · Authors · 2023-08-18
> > > > > >
> > > > > > Thank you very much for your engagement!  We wish to respectfully share what we think of the two points given in your feedback; we appreciate this opportunity to have discussions with you and learn from the interactions.
> > > > > >
> > > > > > We believe structure learning with asymptotic guarantees wouldn’t be possible without appropriate assumptions. Showing the assumptions are necessary and sufficient is usually hard, and we didn’t claim all assumptions in our work are necessary. At the same time, showing that assumptions that are generally sensible and actually sufficient would potentially open the gate to addressing a number of real problems (which we demonstrated experimentally).
> > > > > > Moreover, we are afraid that this criticism, if valid, applies to almost all traditional causal discovery methods: for instance, for the classic method GES[A], we usually assume faithfulness although faithfulness is clearly not necessary — however, formulating necessary conditions is clearly challenging, and even now we don’t have such “necessary” conditions for the asymptotic correctness of GES. We believe this lack of necessary conditions for asymptotic correctness does not diminish the value and elegance of those classic contributions to causal discovery.
> > > > > >
> > > > > > That being said, let us gently add that to address your concerns where three assumptions were mentioned, we discussed in our previous response the necessity of invertibility and faithfulness/minimality, and the role of pure children when interventional data are unavailable and its plausibility for applications, including psychometric studies and image analysis, as suggested by our experiments.
> > > > > >
> > > > > > As expressed above, we highly appreciate this opportunity to exchange opinions with you and learn from your perspective. Please kindly let us know your thoughts, and thank you again for your time and engagement!
> > > > > >
> > > > > >
> > > > > > [A] Optimal Structure Identification With Greedy Search. Chickering et al.

---

> > > > > > > ### Comment · Reviewer_mbLW · 2023-08-18
> > > > > > >
> > > > > > > Thanks for your reply! I am happy to engage in discussions.
> > > > > > >
> > > > > > > I don't doubt the correctness of the theoretical results, although I haven't checked every detail. My point is simply that pure child and faithfulness are two _really strong assumptions_ in causal discovery. Making these two assumptions at the same time lends me to ponder the significance of the results, especially since the discussion of necessity is not rigorous.
> > > > > > >
> > > > > > > Therefore, from my own perspective, I think my assessment is fair. I am still leaning towards accepting, although not in the strongest form.

---

> > > > > > > > ### Author Response · Authors · 2023-08-18
> > > > > > > >
> > > > > > > > Thank you for engaging in the discussion.
> > > > > > > > We agree that some assumptions in causal structure learning might be practically strong. However, for the two specific assumptions used in this contribution, namely, the pure child assumption and faithfulness/minimality, please let us add that they hold true in many real problems.
> > > > > > > >
> > > > > > > > First of all, it was shown that faithfulness-violating systems are Lebesgue measure $0$ with respect to possible causal systems (see, e.g., [A]). Of course, there clearly exist situations where it is violated (see, e.g., [B]), especially on finite data (see, e.g., [C]).
> > > > > > > > On the other hand, structural minimality, into which we weaken faithfulness, is considered necessary for general function classes, as argued in [D] (Section 6.5):
> > > > > > > >
> > > > > > > > > In most model classes, identifiability from observational data is impossible to obtain without causal minimality. We cannot distinguish between $Y := f(X) + N_{Y}$ and $Y := c + N_{Y}$ , for example, if $f$ is allowed to differ from constant $c$ only outside the support of $X$.
> > > > > > > >
> > > > > > > > This example exactly makes a case for the necessity of minimality in our work on nonparametric functions – the edge $X \to Y$ cannot be inferred from data if the function $f(\cdot)$ remains constant over the support of $X$.
> > > > > > > >
> > > > > > > > For the pure children assumption, as we have many directly measured variables in a number of fields, including psychometrics, image analysis, and natural languages, the pure child assumption seems to naturally hold true in such cases. Our experiments are aimed to demonstrate this plausibility in certain cases. We would appreciate it tremendously if you could kindly point us to developments (that we may have overlooked) on latent hierarchical models with observational data that assumes no pure children.
> > > > > > > >
> > > > > > > > Admittedly, assumptions may be annoying. Clearly, in certain situations, our assumptions may be violated. At the same time, they are needed for technical developments with correctness guarantees. But we believe this work is still an essential contribution to the field of structure learning for nonlinear latent hierarchical cases (to the best of our knowledge, this setting has not been explored before, despite its directly real implications). We hope it will inspire better assumptions and better methods. Thank you once again.
> > > > > > > >
> > > > > > > >
> > > > > > > > [A] Causation, Prediction, and Search. Spirtes et al.
> > > > > > > >
> > > > > > > > [B] When to Expect Violations of Causal Faithfulness and Why It Matters. Anderson.
> > > > > > > >
> > > > > > > > [C] Geometry of the faithfulness assumption in causal inference. Uhler et al.
> > > > > > > >
> > > > > > > > [D] Elements of Causal Inference: Foundations and Learning Algorithms. Peters et al.

---

> > > > > > > > > ### Comment · Reviewer_mbLW · 2023-08-19
> > > > > > > > >
> > > > > > > > > Thanks for your reply!
> > > > > > > > >
> > > > > > > > > First of all, the statement that faithfulness-violating systems are Lebesgue measure 0 is _not true_ in general. As far as I know, it is only true for linear models. And as you have said, finite sample estimation typically requires strong faithfulness and strong faithfulness-violating systems are definitely not Lebesgue measure 0.
> > > > > > > > >
> > > > > > > > > Second, I agree that you might need faithfulness or minimality to deal with identifiability under general transformation. I am sorry if I haven't made this clear in my previous post. But my whole point is that you are combining faithfulness/minimality, pure child assumption, and invertibility (assumption 3.3). You can argue why each of them might be needed in your case. However, when you combine multiple assumptions, it's a different story and the significance of the results might be weakened. From this perspective, your arguments on necessity are not super convincing to me.

---

> > > > > > > > > > ### Author Response · Authors · 2023-08-19
> > > > > > > > > >
> > > > > > > > > > Thank you so much for the engagement! We believe we are on the same page that assumptions might be violated in some situations, although they are needed for the purpose of providing asymptotic correctness guarantees. Hope we, as peers, will have the opportunity to discuss more about those assumptions in the future.
> > > > > > > > > >
> > > > > > > > > > Thank you once again for your comments, your engagement, and the great discussions!

---

### Official Review · Reviewer_gbV3 · 2023-07-06

**Soundness:** 3 good
**Presentation:** 3 good
**Contribution:** 3 good
**Rating:** 7
**Confidence:** 3

**Summary:**

This paper addresses the problem of identifying latent variables and causal structures from observational data in the context of nonlinear latent hierarchical causal models. Such models are common in real-world applications involving biological, medical, and unstructured data such as images and languages. The paper presents novel identifiability guarantees and an estimation procedure for both the causal structure and latent variables in these models, under mild assumptions on causal structures and structural functions.

Compared to the previous methods, the main contributions are

1. The author propose the basis model with theoretical guarantees (Theorem 3.2) as a foundation for constructing the identifiability for general nonlinear hierarchical causal model.

2. They show structure identification guarantees for general latent hierarchical models admitting continuous multi-dimensional variables, general nonlinear structural functions, and general graph structures, which go beyond the limitations of previous works that assume linear functions or discrete variables.

3. An estimation method is presented that can asymptotically identify the causal structure and latent variables for nonlinear latent hierarchical models. The authors validate their method on synthetic and real-world datasets.


**Strengths:**

## Originality
The paper presents several original contributions in the context of causal discovery for nonlinear latent hierarchical models. The authors develop a novel identifiability theory (Theorem 3.2) for basis model, which serves as a fundamental criterion for locating and identifying latent variables in general nonlinear hierarchical causal models. This theory relax some of the limitations compared to previous works. Moreover, the paper provides the estimation procedure for general hierarchical structures as well as latent variables up to a invertible mapping.

## Clarity
The paper is logically structured and well-written. I especially appreciated the intuitive explanation behind each assumptions and conditions, making it accessible to general audience. The intuitive explanation of the algorithm is particularly helpful and allowing the reader to capture the main ideas behind the estimation procedure.

## Significance
The paper addresses the causal discovery problem in the context of nonlinear latent hierarchical models.
The proposed identifiability guarantees and estimation procedure relax some of the limitations compared to the previous work, which itself is a great contribution. This paper is likely to have a notable impact on the field of causal discovery and inspire further research in this area.




**Weaknesses:**

This paper is generally well-written, however, I still have some questions:

1. Scalability: The author admits the computational limitations. I still think the discussion on the computational complexity should added. Also it is very helpful to report the wall clock comparison to the other baselines regarding the high dimensional problem.

2. Joint invertible mapping: One of the basic assumptions is that the latent and observational variables are jointly invertible. I wonder if this is not satisfied, you mentioned that the identifiability cannot be established. Could you point out the relevant references? Also, in the experiments, why you use MLP as the function $g$? In general MLP is not invertible, for example, we can construct an MLP that map the $\mathbb{R}^n$ to a fixed point. In this case, you you cannot recover the original latent information.


**Questions:**

The questions are mentioned in the weakness section.

**Limitations:**

The author included a discussion on the limitations regarding the computational complexity. But I think more statics like wall clock time and detailed discussion on the complexity should be added.

---

> ### Author Rebuttal · Authors · 2023-08-10
>
>
> Thank you for your encouraging comments and valuable suggestions! We will include your feedback and the discussions in our revision. Below are our responses to individual questions.
>
> **Q1: Computation complexity and wall-clock times.**
>
> Thank you for the wonderful suggestion – this will help future readers better understand the complexity aspect! We will include the following discussion on computational complexity in our work:
>
> "The dominant complexity term in our proposed algorithm is the deep learning training which is of a complexity $O( R * n )$ where $R$ is the number of levels in the hierarchical model, and n is the number of observed variables. In contrast, recent works on linear hierarchical models provide algorithms with complexity $O( R * (1+n)^{p+1} )$ [15] and $O(R * n!)$ [35] and $p$ is the largest number of variables that share children. Our algorithm achieves the lowest complexity in terms of the algorithmic complexity.  As we allow for multi-dimensional variables, the dimensionality of each variable also contributes to the overall wall-clock time through the deep learning model dimension, which is a hyperparameter for the experimenter. This cost can grow significantly for high-dimensional datasets like ImageNet."
>
> In the attached PDF file, we report detailed wall-clock times for all experiments. We note that the wall-clock time can vary significantly depending on the CPU availability, as we run kernel regression on CPUs for $R^{2}$ score computation. Most of our experiments were launched on a group server and experienced contending CPU usage to certain extents.
>
> Our basis model estimation takes roughly the same time as the baseline method in Table 1.
> As no existing causal discovery approaches are designed for nonlinear hierarchical models with multi-dimensional variables (to the best of our knowledge), we cannot find a baseline to compare with on this aspect.
>
> **Q2: Joint invertible mapping and MLPs.**
>
> Great questions! We’d like to note that, to the best of our knowledge, existing latent variable identification literature universally adopts the invertibility (or, more broadly, injectiveness) assumption. Please see [18,19,21,24,36,37].
>
> The fundamental issue is as follows. The observed variables are generated as a function of latent variables, i.e., $ x := g(z_{1}, ..., z_{i}, ..., z_{n})$ where $x$ is the observed variable, $z_{i}$ is the latent variable, and $g$ is the function.
> Identification results typically entail that we recover/identify each latent variable $z_{i}$ from the observed variable $x$, i.e., $z_{i} = f_{ z_{i} } (x)$ via a learned function $ f_{z_{i}} $.
> To express each $z_{i}$ as a function of $x$, it is necessary that $x$ preserves all original information of $z_{1}$, ..., $z_{n}$, which is equivalent to the mapping $g$ is invertible (injective).
>
> Regarding MLPs, we fully agree that general MLPs can be non-invertible.
> In our experiments, we follow the implementation of prior work [18,19,21] to adopt leaky-ReLU and dimension-preserving MLP layer weights with large condition numbers to facilitate invertibility. We will expand on this in our paper to make it clearer -- thank you so much for the suggestion!
>
>
>
> Please let us know if you have further concerns – thank you!

---

> ### Comment · Reviewer_gbV3 · 2023-08-16
>
> Thanks for the authors' reply. It addresses my concerns. I will keep my original score.

---

> > ### Author Response · Authors · 2023-08-16
> >
> > Thank you so much for the time, effort, and acknowledgment of our work!

---

### Official Review · Reviewer_wEjX · 2023-07-06

**Soundness:** 3 good
**Presentation:** 2 fair
**Contribution:** 3 good
**Rating:** 5
**Confidence:** 4

**Summary:**

This paper presents an identification strategy that allows the unique reconstruction of a latent hierarchical model, including both the graphical structure and (up to invertible transformations) the values of the latent variables. Assumptions include faithfulness, some structural assumptions (weak compared to related work), and other more technical assumptions. The algorithm is positively evaluated on various datasets.

**Strengths:**

* Identifying latent structure (and even the values of latent variables themselves) is very powerful and could be used in many applications.

* There are only weak assumptions on the graphical structure.

* The experimental results look very good.

**Weaknesses:**

* Differentiability is assumed, but not mentioned explicitly (another assumption refers to Jacobians). The strength of a continuity/differentiability assumption is compounded by the assumption of invertibility.

* I find it hard to assess how strong the subspace span condition is, in part because it is not written up clearly; see questions below.

* Footnote 1 / appendix C.2 address the case where some non-leaf variables are observed. For this case, I think the structural assumptions become quite strong, so I do not consider this a strong part of the paper's contribution.

**Questions:**

* line 97-99, "a latent variable with no pure children can be merged into its children without affecting the overall generating process": this merging appears to not be possible for the middle z in $x \leftarrow z \rightarrow x \leftarrow z \rightarrow x \leftarrow z \rightarrow x$. Could you explain what you mean exactly?

* Assumption 3.1(ii) and text before it: The definition of T depends on c, but for what value? Further, how is T quantified over in assumption 3.1(ii)?

* If $g$ is linear, the subspace span condition (ii) will not hold (except in 1-dimensional cases, I suppose). Can this be addressed by something like "restructure latent variables by merging redundant components" (line 156)?

  * Related: in line 200-202, the identity function doesn't satisfy assumption 3.1(ii) if $d_z > 1$

* Assumption 3.3(ii): Is the independence in 1) automatically satisfied if  $A_z$ and $X$ have nonempty intersection? (Please make this explicit or rewrite to avoid this case.) I suppose condition 2) refers to the induced subgraph of $A_z$, as a set of node does not strictly speaking contain any paths. Is there a minimum length to these paths? A 0-length path always exists, and if you mean a 1-length path, you could simply ask if there is an arrow between two nodes in $A_z$.

* Assumption 3.3(iii): The "function between each latent variable $z$ and each of its children $z'$ (I assume separately for each $z'$?) also depends on other parents of $z'$. Should this assumption hold for some/most/all of the values of those parents?

* Algorithm line 17: remove variables independent of what? In the example, the only role of this line appears to be to clear A if it contains just the root. What else should this line do?

Things to clarify:

* latent variable identification: I'd recommend mentioning early on that this is up to transformations. Also, when this is first mentioned on the top of page 3, the objective is impossible to obtain without the assumption 3.3(i), which appears quite a bit later.

* Condition 2.3 (ii): I recommend to give a definition for "siblings" here

* In the algorithm, how do you detect equivalence of two constructed latent variables? (Mutual perfect predictability I suppose?)

* Algorithm line 9: There needs to be "\ {$\hat{z}$} after $P$.

* Line 352-353: the part after "whereas" doesn't match the results

Textual comments:

* line 47 "variable variables"

* line 171: "descents" -> "descendants"

* line 211-212: "$v_2$ always contains" -> "there always exists a $v_2$ [$\neq v_1$]"

* line 248 & 337: "an variable" -> "a"

* line 266 & 524: "an one-to-one mapping" -> "a"

* line 523: "proof" should instead refer to theorem / assumptions

**Limitations:**

The paper has many assumptions, which are hard to interpret. While some effort is made to discuss most of these (exceptions: see weaknesses), this discussion leaves some questions (see above).

---

> ### Author Rebuttal · Authors · 2023-08-10
>
> We deeply appreciate your thorough reading and valuable insights! Below, we provide individual responses to your comments.
>
> **Q1: Differentiability is not explicit.**
>
> Thank you for pointing this out. This condition should definitely be given, and we have made this assumption explicit in our revision. We'd like to note that the differentiability assumption and invertibility are adopted in almost all continuous latent-variable identification papers [18,19,21,24,36,37].
>
> **Q2: The observed non-leaf case is not a strong contribution.**
>
> We agree with you – Footnote 1 is only meant to illustrate that our theorem applies naturally to models with observed non-leaf variables beyond our main discussion, where all non-leaf variables are latent.
>
> **Q3: ``Line 97-99,..., the merging seems not possible….’’**
>
> Thank you for raising the point. We meant to describe scenarios where latent variables are unidentifiable for lack of unique footprints. For instance, if two latent variables $z_{1}$ and $z_{2}$ share the same set of children $X$, i.e., $ z_{1} \to X$ and $z_{2} \to X$, then the two latent variables cannot be identified without further assumptions, while pure children would help in this case.
>
> In light of your remark, we will replace the sentence in lines 97-99 “for instance, …” with “in a toy graph $\{z_{1}, z_{2}} \to X$, two latent variables $z_{1}$ and $z_{2}$ cannot be distinguished without further assumptions.”
>
> **Q4: The definition of $T$.**
>
> Thank you for the question. In lines 135-136, we define $T$ as a fixed matrix that shares the support (defined in lines 130-132) of the matrix-valued function $T(c)$. Therefore, $T$ and $T(c)$ refer to two different objects, and $T$ is not dependent on $c$. This assumption is adopted in prior work [A]. Sorry for the confusion – we will replace the fixed matrix $T$ with $Q$ in the revision.
>
> **Q5: ``If $g$ is linear, the subspace span condition (ii) will not hold….’’**
>
> Great observation! The subspace span assumption (ii) stipulates that the generating function's Jacobian should vary sufficiently within its support and is specialized for nonlinear functions, as discussed in [A,B]. Prior work [A] derives counterpart assumptions for the linear case (see Proposition 1, Assumption ii in [A]). These conditions can be adapted to our setup for the linear case. In addition, Prior work [B] supplies an explanation of the subspace span assumption (ii) and an example of functions that satisfies such an assumption (see Section 2.4.1. in [B]). In light of your question, we will include this discussion and the example in our revision to make our work thorough.
>
> Regarding the identity function, we note that its Jacobian matrix $G$ is an identity matrix. The resultant subspace $R_{ G_{ i , :} }^{d_{c}}$ is only a 1-d subspace (see the subspace definition in line 128) in $R^{d_{c}}$, because row $i$ only has one nonzero element $G_{i,i} = 1$. Consequently, Assumption 3.1 (ii) is met even if $d_{c} > 1$.
>
> **Q6: Assumption 3.3 (ii) and path lengths.**
>
> Great question! When $A_{z} $ and $ X$ have a nonempty intersection, the independence condition 1) does not necessarily hold. Take a toy graph $z \to x_{1}$, $z \to x_{2}$ and set $A_{z} = \\{ x_{1} \\}$. We have $ X \cap A_{z} = \\{ x_{1}\\} \neq \emptyset$ and $ z \not\perp X | A_{z} $ because the path $z \to x_{2}$ is unblocked.
>
> As you correctly identified, we refer to the subgraph induced by variables in $A_{z}$, and the paths refer to paths with nonzero lengths. Thank you for making this distinction, and we will explicitly make this point in our manuscript.
>
> **Q7: Assumption 3.3 (iii).**
>
> Great question! As this assumption is made on the support of the Jacobian matrix (see lines 130-132), it is satisfied as long as the component $z_{j}$ has a nonzero influence (i.e., partial derivative) on its child's component $z'_{i}$ at *some* values of parents of $z’$.
>
> **Q8: Algorithm line 17.**
>
> Algorithm line 17 removes variable $a \in A$ if $a$ is independent of all other variables in $A$, i.e., the complement set $ A  \backslash \\{ a \\} $.
> Some root variables may enter $A$ before the other root variables for graphs with multiple root variables. Line 17 removes these identified root variables from the active set $A$ before the iteration. One instance would be $z_{1} \to \\{ x_{1}, x_{2}, x_{3}\\}$, $z_{2} \to \\{ x_{3}, x_{4}, x_{5}\\}$, $z_{3} \to \\{ x_{6}, x_{7}\\}$, and $ z_{4} \to \\{ z_{2}, z_{3} \\} $. In this case, one root variable $z_{1}$ would enter $A$ before the other root variable $z_{4}$ and would get cleared earlier. We will improve this in the manuscript, thanks to your feedback.
>
> **Q9: Mention “up to transformations” early on and the definition of siblings.**
>
> Wonderful suggestions -- this will definitely help us improve the readability! We will mention "up to invertible transformations" in the introduction and abstract, and we will add a footnote for the objective to indicate that assumptions such as Assumption 3.3 (i) are necessary. We will include the definition of siblings right before introducing Condition 2.3.
>
> **Q10: Equivalence detection.**
>
> You are totally right! We use mutual predictions to detect the equivalence between two estimated variables. In our implementation, this is done by kernel regression (line 314).
>
> **Q11: Algorithm line 9.**
>
> Thank you for pointing this out! We will correct it as you suggested.
>
> **Q12: ``Line 352, …, does not match…’’.**
>
> Thank you! We will replace "closely" with "remotely".
>
> **Q13: Typos.**
>
> We are grateful for all the listed typos, and we will correct them in the paper. Thank you for your time and effort!
>
> **References:**
>
> [A] On the Identifiability of Nonlinear ICA: Sparsity and Beyond. Zheng et al.
>
> [B] Disentanglement via Mechanism Sparsity Regularization: A New Principle for Nonlinear ICA. Lachapelle et al.
>
> Please let us know if you have further concerns, and please consider raising the score if we have cleared existing concerns  – thank you so much!

---

> > ### Comment · Reviewer_wEjX · 2023-08-17
> >
> > Thank you for your replies, which have addressed most of my concerns.
> >
> > Regarding Q4: It might be better to leave the name $T$ as it is. My confusion stems from the fact that $T$ is defined in terms of $T(c)$, but it was not clear to me why $T(c)$'s support should be the same for general $c$, because I didn't recognize this construction.
> >
> > Knowing that these assumptions have appeared in the prior work mentioned in the rebuttal, somewhat alleviates my concerns about the clarity of the assumptions' exposition in this paper. While I would have preferred to have known these references during the review, I am raising my score from 4 to 5.

---

> > > ### Author Response · Authors · 2023-08-17
> > >
> > > Thank you for your valuable feedback and thoughtfulness. We’ve improved the definition and included the references in our revision to make the construction transparent to the readers.
> > >
> > > The revised texts are as follows:
> > >
> > > Line 135: “We denote Jacobian matrices of $g$ and $\hat{g}$ as $J_{g}$, and their supports as $G$ and $\hat{G}$, respectively.
> > > Further, we denote as $T$ a fixed matrix that has the same support as the matrix-valued function $J_{g}^{-1} (c) J_{\hat{g}} (\hat{c})$.”
> > >
> > > Line 153: “Assumption 3.1 (ii), which is given in prior work [A,B], guarantees that the influence of $z$ changes adequately across its domain.”
> > >
> > > We were wondering if all your concerns had been properly addressed – please kindly let us know if you have any further concerns. Many thanks for your time and dedication!

---

> > > > ### Author Response · Authors · 2023-08-21
> > > >
> > > > Dear reviewer wEjX,
> > > >
> > > > We thank you once again for your time and efforts.  Since the discussion period will end in one hour, we will be online waiting to see whether your previous concern was properly addressed. We would highly appreciate it if you could take into account our response when updating the rating and having discussions with AC and other reviewers.
> > > >
> > > > Thanks a lot,
> > > >
> > > > Authors of #2805

---

### Author Rebuttal · Authors · 2023-08-10

We thank all reviewers for their valuable feedback and dedicated time! We are encouraged that they find our theoretical contribution significant/substantive (wEjX, gbV3, mbLW, jcJm) and well-supported by experimental results (wEjX, mbLW). We address the individual comments in separate responses and will incorporate the reviewers’ suggestions in our revision.

---

### Decision · Program_Chairs · 2023-09-21

**Decision:**

Accept (poster)

**Comment:**

All reviewers except one (jcJm) argued for accepting the paper. For this reviewer their main conerns were on the related work, paper clarity, and experiments. Their recommendations were to (a) include a discussion on identifying latent causal graphs from interventional data, (b) clarify whether deterministic or random functions are assumed in the graph, (c) more gradually introduce algorithm 1, (d) clarify if there is a contradiction between theorem 4.3 and condition 2.3, (e) improve description of experiments, (f) add additional experiments where sample size is varied. The authors responded convincingly to each of these points. If the reviewer had continued to engage in the discussion they would have likely voted to accept (as they mention in a response to the authors). Further, the authors did a stellar job responding to the concerns of other reviewers. For these reasons, I vote to accept. Authors: you’ve already made improvements to respond to reviewer changes, if you could double check their comments for any recommendation you may have missed on accident that would be great! The paper will make a nice contribution to the conference!